# A conserved expression signature predicts growth rate and reveals cell & lineage-specific differences

Zhisheng Jiang[1]⊘, Serena F. Generoso[2]⊘, Marta Badia[3], Bernhard Payer[2,3]*, Lucas B. Carey[1,3]*

**1** Center for Quantitative Biology and Peking-Tsinghua Center for Life Sciences, Academy for Advanced Interdisciplinary Studies, Peking University, Beijing, China, **2** Centre for Genomic Regulation (CRG), The Barcelona Institute of Science and Technology, Barcelona, Spain, **3** Universitat Pompeu Fabra (UPF), Barcelona, Spain

⊘ These authors contributed equally to this work.
\* bernhard.payer@crg.eu (BP); lucas.carey@pku.edu.cn (LBC)

**Data Availability Statement:** All RNA-seq files created in this study are available from the GEO database (accession number GSE139594).

## Abstract

Isogenic cells cultured together show heterogeneity in their proliferation rate. To determine the differences between fast and slow-proliferating cells, we developed a method to sort cells by proliferation rate, and performed RNA-seq on slow and fast proliferating subpopulations of pluripotent mouse embryonic stem cells (mESCs) and mouse fibroblasts. We found that slowly proliferating mESCs have a more naïve pluripotent character. We identified an evolutionarily conserved proliferation-correlated transcriptomic signature that is common to all eukaryotes: fast cells have higher expression of genes for protein synthesis and protein degradation. This signature accurately predicted growth rate in yeast and cancer cells, and identified lineage-specific proliferation dynamics during development, using *C. elegans* scRNA-seq data. In contrast, sorting by mitochondria membrane potential revealed a highly cell-type specific mitochondria-state related transcriptome. mESCs with hyperpolarized mitochondria are fast proliferating, while the opposite is true for fibroblasts. The mitochondrial electron transport chain inhibitor antimycin affected slow and fast subpopulations differently. While a major transcriptional-signature associated with cell-to-cell heterogeneity in proliferation is conserved, the metabolic and energetic dependency of cell proliferation is cell-type specific.

## Author summary

By performing RNA sequencing on cells sorted by their proliferation rate, this study identifies a gene expression signature capable of predicting proliferation rates in diverse eukaryotic cell types and species. This signature, applied to single-cell RNA sequencing data from embryos of the roundworm *C. elegans*, reveals lineage-specific proliferation differences during development. In contrast to the universality of the proliferation signature, mitochondria and metabolism related genes show a high degree of cell-type specificity; mouse pluripotent stem cells (mESCs) and differentiated cells (fibroblasts) exhibit

**Funding:** This work has been funded by the Spanish Ministry of Science, Innovation and Universities (BFU2014-55275-P and BFU2017-88407-P to B.P. and BFU2015-68351-P to L.B.C.), the AXA Research Fund and the Agencia de Gestio d'Ajuts Universitaris i de Recerca (AGAUR, 2017 SGR 346 to B.P. and 2014 SGR 0974 & 2017 SGR 1054 to L.B.C.), the National Natural Science Foundation of China (31950410537 to L.B.C.). We would like to thank the Spanish Ministry of Economy, Industry and Competitiveness (MEIC) to the EMBL partnership, to the 'Centro de Excelencia Severo Ochoa', and the Unidad de Excelencia María de Maeztu, funded by the MINECO (MDM-2014-0370). We also acknowledge the support of the CERCA Programme of the Generalitat de Catalunya. L.B.C. was supported by funding from Peking University and from the Peking-Tsinghua Center for Life Sciences, and from the Research Fund for International Young Scientists (National Natural Science Foundation of China). The funders had no role in study design, data collection and analysis, decision to publish, or preparation of the manuscript.

**Competing interests:** The authors have declared that no competing interests exist.

opposite relations between mitochondria state and proliferation. Furthermore, we identified a slow proliferating subpopulation of mESCs with higher expression of pluripotency genes. Finally, we show that fast and slow proliferating subpopulations are differentially sensitive to mitochondria inhibitory drugs in different cell types.

Highlights:

1. A FACS-based method to determine the transcriptomes of fast and slow proliferating subpopulations.

2. A universal proliferation-correlated transcriptional signature indicates high protein synthesis and degradation in fast proliferating cells across cell types and species.

3. Applied to scRNA-seq, the expression signature predicts the global proliferation slow-down during *C. elegans* development.

4. Mitochondria membrane potential predicts proliferation rate in a cell-type specific manner, with ETC complex III inhibitor having distinct effects on fibroblasts vs mESCs.

## Introduction

Rates of cell growth and division vary greatly, even among isogenic cells of a single cell-type, cultured in the same optimal environment [1]. Cell-to-cell heterogeneity in proliferation rate has important consequences for population survival in bacterial antibiotic resistance, stress resistance in budding yeast, and chemo-resistance in cancer [2–10]. A recent study has demonstrated semi-heritable cell-to-cell heterogeneity in gene expression in mammalian cells, which is associated with drug resistance in cancer [11] and time-lapse fluorescence microscopy has shown that cell-to-cell variability in the expression of some genes, such as *p53* and *p21*, is associated with cell-to-cell variability in proliferation and survival [1,12]. While microscopy can identify dynamic relationships between gene expression and cell fate, it is limited to measurements of one or two genes per cell. Single-cell RNA sequencing measures transcriptome-level heterogeneity but does not directly link this to cell-biological heterogeneity in organelle state, or dynamic heterogeneity in proliferation or drug resistance. Transcriptome-level approaches for understanding within-population cell-to-cell heterogeneity in proliferation and other dynamic processes are lacking. While the presence of intrapopulation variation in proliferation, transcriptome, and organelle-state in both steady-state and in differentiation populations is well established, the relationship among the three remains unclear.

One possibility is that the proliferation-correlated gene expression program is the same, regardless of if one looks at interpopulation variation due to genetic or environmental differences, or intrapopulation heterogeneity due to epigenetic differences and expression noise. However, in the budding yeast *Saccharomyces cerevisiae*, the expression program of intrapopulation heterogeneity in proliferation rate only partially resembles that of cells growing at different rates due to genetic or environmental perturbations[8]. The relation between gene expression and proliferation rate is much less well studied in mammalian cells.

In yeast, in tumors, and in organs, genetic, environmental and developmental changes cause changes in proliferation rate, and changes in the expression of hundreds or thousands of genes [13–17]. Unsurprisingly, many of the genes for which changes in expression are associated with changes in proliferation rate are associated with adverse clinical outcomes in cancer and with antibiotic and antifungal resistance [18,19]. Within a population of microbes, and

within a single multicellular organism, the correct balance of proliferation states and rates is essential. Yet measuring this heterogeneity is difficult, and without such data, understanding the consequences of this heterogeneity is impossible.

Gene expression is associated with phenotype, but mRNAs themselves do not often directly cause phenotypes. Instead, they serve as markers for cell-biological differences between cells. Phenotypes are mostly driven by larger cell-biological differences between cells, such as differences in metabolic state. Cell-to-cell heterogeneity in mitochondria state has been linked to differences in transcription rates, growth rates, proliferation and developmental trajectories [20–22]. Both cancer cells and pluripotent stem cells have atypical metabolisms and use glycolysis to produce much of their ATP, instead of the mitochondria-based oxidative phosphorylation, which is the predominant form of ATP-generation in differentiated cells [23]. It is unknown if this inter-population variation in proliferation, transcriptome, and mitochondria extends to intra-population variation among single cells within a single isogenic population.

Pluripotent stem cells exist in various states, such as naïve or primed, based on culture conditions and embryonic origin [24]. Mouse embryonic stem cells (mESCs) reflect the naïve pluripotency state of the blastocyst epiblast and can be cultured in either serum+LIF or 2i+LIF conditions, the latter involving inhibitors of FGF/ERK and GSK3 pathways. Culture in 2i+LIF conditions promotes a ground state more closely mirroring the *in vivo* situation with reduced heterogeneity in pluripotency gene expression and different cell cycle profile when compared to cells grown in serum+LIF [25–27]. Nevertheless, even in 2i+LIF conditions, mESCs display a certain amount of cell-to-cell heterogeneity [28,29] and it is unclear, how this relates to heterogeneity in differentiated cell types when it comes to gene expression and its link to proliferation rate.

To understand the relation between intra-population transcriptome heterogeneity and heterogeneity in proliferation, we developed a FACS-based method to sort cells by proliferation rate. We applied this method to mouse immortalized fibroblasts and mESCs and performed RNA-seq on fast, medium and slow proliferating cell sub-populations. We identified a "proliferation signature", mostly consisting of ribosome-biogenesis (protein synthesis) and proteasome-related (protein degradation) genes that are highly expressed in fast proliferating fibroblasts and ESCs. Moreover, the proliferation signature is conserved across cell-type and species, from yeast to cancer cells, allowing us to predict the relative proliferation rate from the transcriptome. We used this gene expression signature to predict proliferation rates in single cells from scRNA-seq data of *C. elegans* development. Unlike previous models to predict growth rate from gene expression [30], this model has no free parameters other than the set of genes, and does not suffer from overfitting–it can predict differences in growth rate in yeast, cancer cells and *C. elegans*, despite no data from either species going into the initial model. When applied to scRNA-seq data from developing *C. elegans*, this model identified a global slowdown in proliferation rate during development, with lineage-specific exceptions where some lineages maintain constant proliferation scores, and others even increase proliferation rate. In contrast to the universality of this main transcriptional signature, many mitochondria-related genes were upregulated in fast proliferating fibroblasts, yet down-regulated in fast-proliferating mESCs. Consistent with this, we found that a high mitochondria membrane potential is indicative of slow proliferating fibroblasts, while in mESCs this is a property of fast proliferating cells. And the mitochondrial electron transport chain complex III inhibitor antimycin treatment causes opposite effects on the proliferation of fibroblasts and ESCs. Fast, but not slow proliferating fibroblasts are particularly sensitive to the ATP synthase inhibitor oligomycin. Taken together, these results show the existence of a core protein-synthesis and protein-degradation expression program that is conserved across cell types and species, from yeast to mice, and a metabolic and energy-production program that

is highly cell-type specific, with cell-type and proliferation-rate specific consequences on the effects of mitochondria inhibitors.

## Results

### A method to sort single mammalian cells by semi-heritable cell-to-cell heterogeneity in proliferation rate

To understand the causes and consequences of intrapopulation cell-to-cell heterogeneity in proliferation rate in mammalian cells we developed a method for sorting single mammalian cells by their proliferation rate (**Figs 1 and S1**). The cell-permeable dye carboxyfluorescein succinimidyl ester (CFSE) covalently binds free amines within cells, thus staining most intracellular proteins at lysine residues. In cell types that divide symmetrically, such as embryonic stem cells and immortalized fibroblasts [31], the equal dilution of CFSE into the two daughter cells enables counting of the number of divisions that each cell has undergone. This method is commonly used to differentiate proliferating from nonproliferating cells, and to count discrete numbers of cell division, such as in the study of T- and B-cell proliferation following antigen stimulation [32]. To eliminate confounding effects due to differences in initial staining we used fluorescence-activated cell sorting (FACS) to obtain an initially homogeneous cell population of cells with identical CFSE signals (**Figs 1, S1A and S1B**). Thus, the initial CFSE signal is independent of initial cell-to-cell variation in dye uptake or protein content, as the initial distribution is determined by the FACS gate. $CFSE_{CFR2}$ conjugates are stable and unable to exit the cell [33]; the dye signal is stable for over eight weeks in non-dividing lymphocytes [34]. The measured CFSE signal should be relatively insensitive to cell-to-cell variation in protein degradation. We cultured this sorted starting cell population for several generations, during which time the CFSE signal decreases with each cell division (**Fig 1B**). Consistent with the decrease in CFSE being mostly due to cell division, the population-level doubling time of each cell type can be calculated based on the decrease in CFSE signal over time (**Fig 1C and 1D**), and these doubling times (19–21 hours for fibroblasts and 10–12 hours for mESCs) are consistent with those reported by other methods [35,36].

To test if the intrapopulation heterogeneity in CFSE that develops after a few doublings corresponds to intrapopulation heterogeneity in proliferation rates, we stained cells with CFSE, isolated a homogenous population by FACS, grew ESCs or fibroblasts for 24h or 48h respectively, and used FACS to isolate the 20% of cells with the highest and lowest CFSE signal, and measured both viability and the fraction of cells in S phase (**S1C and S1D Fig**). We found that fast proliferating (low CFSE) subpopulations maintain higher proliferation rates for at least three days (**Figs 1E, 1F, S1E and S1F**), and found no differences in viability between CFSE subpopulations (**S1 Table**). Thus, intrapopulation differences in CFSE correspond to semi-heritable differences in proliferation rates.

To identify genes whose expression is positively or negatively correlated with proliferation rate within a single population we grew fibroblasts in MEF (mouse embryonic fibroblast) medium and mouse ESCs in pluripotent ground-state promoting 2i+LIF medium [37], stained cells with CFSE, performed the initial sort to isolate cells with the same CFSE signal, and then grew fibroblasts for five days, and ESCs for three days. We then used FACS to isolate cells with high, medium, and low CFSE signal, and performed RNA-seq on each sub-population (**Fig 1G**).

### Slow-proliferating ESCs are of more naïve pluripotent character than fast-proliferating ESCs

Embryonic stem cells exhibit cell-to-cell heterogeneity in the expression of naïve pluripotency marker genes such as *Nanog*, *Stella* (*Dppa3*) or *Rex1* (*Zfp42*) [38–40]. Although this

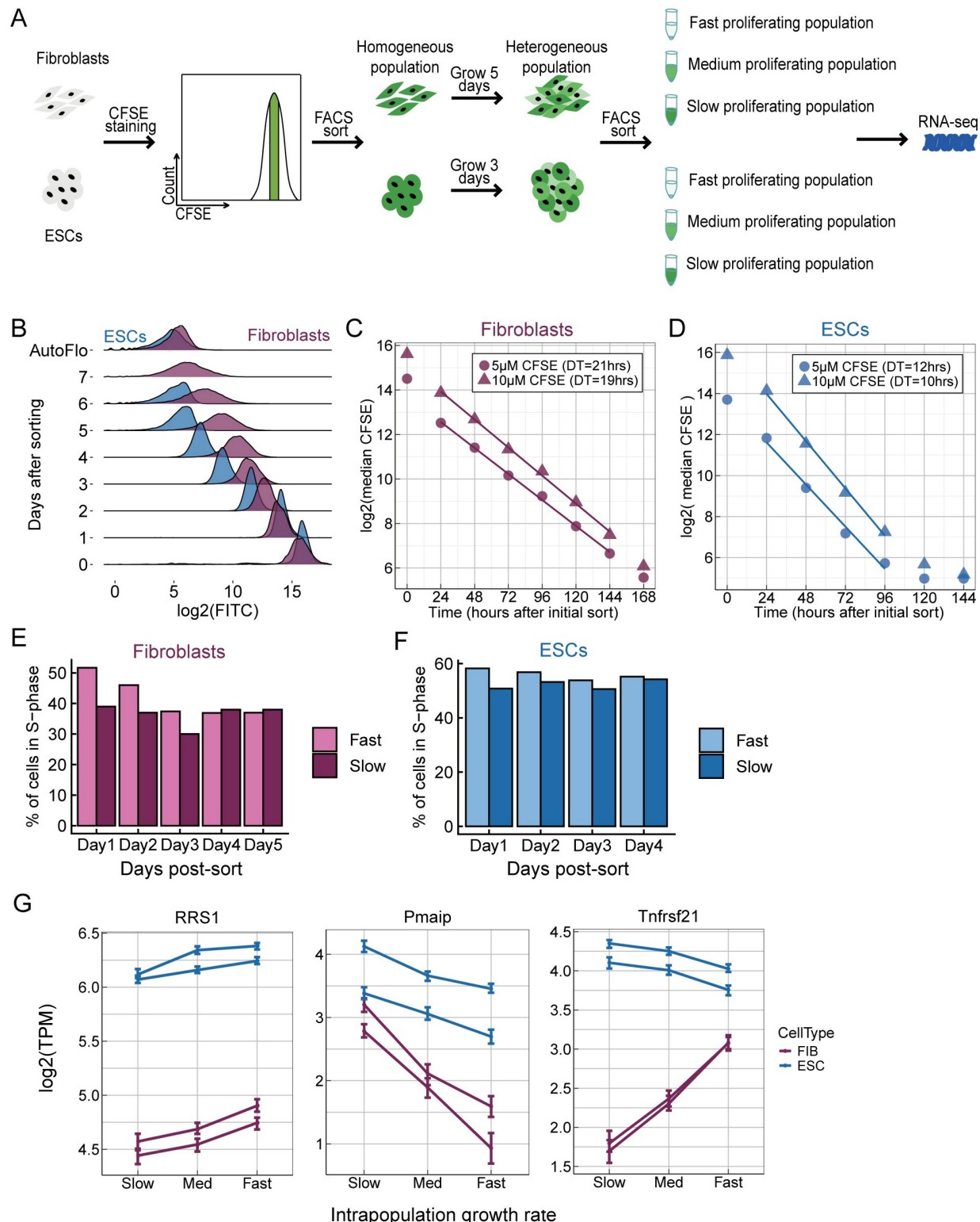

**Fig 1. A CFSE-based method to sort mammalian cells by proliferation rate.** (A) Cells were stained with CFSE and a subpopulation of cells with identical CFSE levels was collected by FACS. Growth for several generations resulted in a heterogeneous cell population with a broad CFSE distribution, and cells with high, medium, and low CFSE signal (slow, medium and fast proliferation, respectively) were sorted by FACS for RNA-sequencing. (B) The change in the CFSE distribution over time, for fibroblasts and ESCs. (C, D) The population-level doubling time can be calculated by fitting a line to the median of the log2(CFSE) signal. We discard data from time 0, cells immediately after the sort, because the CFSE

signal decreases in the initial hours, even in the absence of cell division, likely due to efflux pumps. **(E, F)** Bromodeoxyuridine (BrdU) was used to measure the % of cells in S-phase for FACS-sorted fast and slow proliferating subpopulations. Fibroblasts: 4 replicates, p = 0.0002441. ESCs: 3 replicates for ESCs, p = 0.001953. p-values are for binomial tests across all biological replicates that the two populations have the same percentage of cells in S-phase. **(G)** Examples of genes whose expression positively or negatively correlated with proliferation rate. Each line is one biological replicate, and the error bars are 95% confidence intervals for each expression value.

heterogeneity is most apparent in ESCs cultured in serum+LIF, even when cultured in ground state-pluripotency-promoting 2i+LIF conditions, the sub-population of ESCs with low NANOG-levels displays a propensity for lineage-priming and differentiation [28,29]. To determine if cell-to-cell variation in proliferation rate was caused by a sub-population of mESCs initiating a differentiation program, we determined the fold-change in expression between slow and fast proliferating sub-populations for a set of genes that are upregulated during lineage commitment (see "Differential expression of pluripotency. . ." in Materials and Methods). We found no consistent enrichment of these differentiation genes in fast versus slow proliferating cells, as they could be found to be expressed in either population (**Fig 2A**). However, the slow proliferating subpopulation did have higher expression of genes that are upregulated in naïve pluripotent cells, and in 2-cell(2C)-like state stem cells (**Fig 2B and 2C**), suggesting that slow proliferating mESCs are in a more naïve pluripotent cell state than their fast proliferating counterparts.

## Identification of biological processes correlated with proliferation rate that are conserved across cell-types and species, and within single populations

To identify functional groups of genes that are differentially expressed between fast and slow proliferating cells within a single population we performed gene set enrichment analysis (GSEA)[41,42] (**Figs 3A, 3B, S2A and S2B**) on mRNA-seq data from fast and slow proliferating subpopulations. We found that in both fibroblasts and ESCs, as well as for intrapopulation variability in budding yeast FACS-sorted by proliferation rate (data from van Dijk et al. [8]), genes involved in ribosome-biogenesis and the proteasome are more highly expressed in fast proliferating cells (**Fig 3C and 3D and S2 Table**). High expression of ribosomal genes is a common signature for fast proliferating cells [13,43,44], and cancer cells often exhibit high

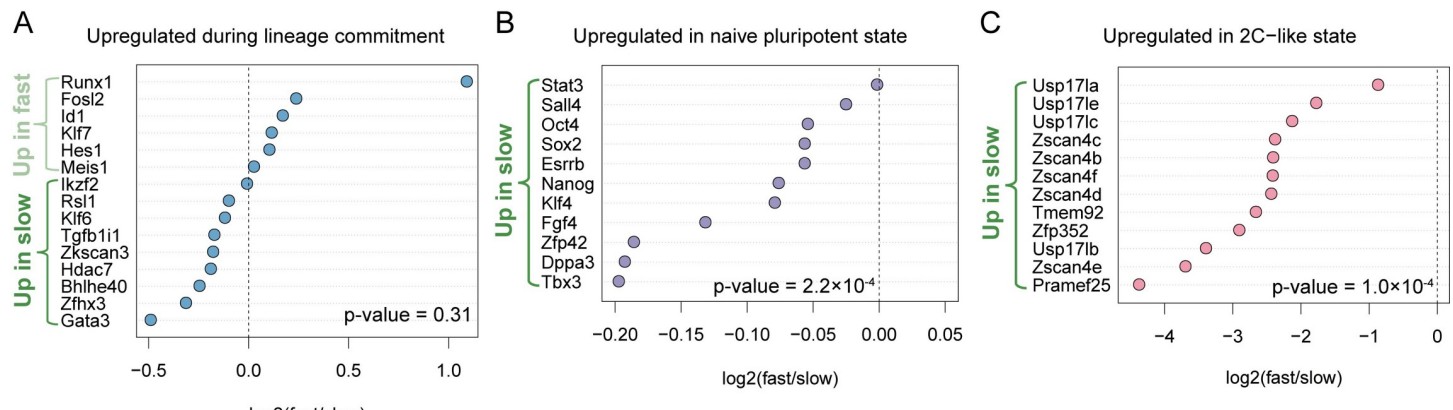

**Fig 2. Slow-proliferating ESCs display a more naive pluripotent stemness character than fast-proliferating ESCs. (A)** Comparison of lineage commitment-related gene expression between fast and slow proliferating subpopulations. **(B)** Comparison of pluripotency-associated gene expression between fast and slow proliferating subpopulations. **(C)** Comparison of 2C-like state markers expression between fast proliferating subpopulation and slow proliferating sub-population. Dashed lines separate genes expressed preferentially in slow- (left of dashed line) or in fast-proliferating (right of dashed line) ESCs. P-values are from binomial tests, testing if genes are more often highly expressed in slow cells than would be expected by chance (53.5% of all genes are more highly expressed in slow cells).

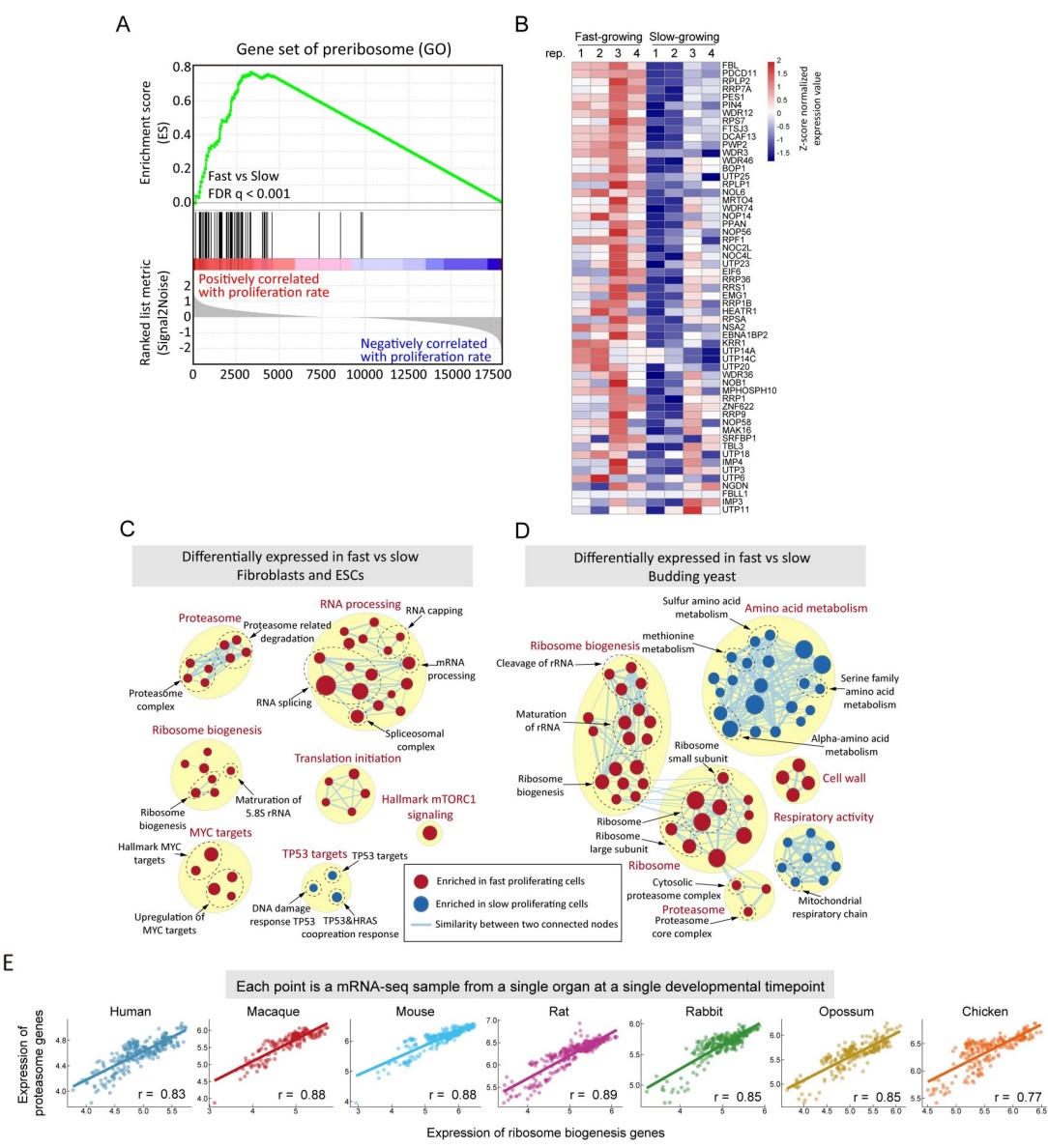

**Fig 3. Functional pathways for which cell-to-cell heterogeneity in expression correlates with proliferation rate across cell types and species. (A)** In Gene Set Enrichment Analysis, genes are sorted by their fast/slow expression value (left panel, bottom), and each gene is represented by a single black line (left panel middle). The enrichment score is calculated as follows: for each gene not in the GO preribosome gene set, the value of the green line decreases, and for each gene in the gene set, the value of the green line increases. The ES score will be near zero if the genes in a gene set are randomly distributed across the sorted list of genes, positive if most genes are to the left, and negative if most genes are to the right. **(B)** The heatmap (right panel) shows the expression (z-scored read counts) of preribosome genes in fibroblasts across four biological replicates of the CFSE sorting experiment. **(C)** Gene sets enriched (FDR<0.1) in both fibroblasts and ESCs were mapped as a network of gene sets (nodes) related by mutual overlap (edges), where the color (red or blue) indicates if the gene set is more highly expressed in fast (red) or slow (blue) proliferating cells. Node size is proportional to the total number of genes in each set and edge thickness represents the number of overlapping genes between sets. **(D)** GSEA results (FDR<0.1) of *S. cerevisiae* [8] that sorted by cell-to-cell heterogeneity in proliferation rate. **(E)** Pearson correlations of mean expression (average of log2(TPM+1)) of ribosome biogenesis genes vs proteasome genes across organ developmental time courses in seven species (see also **S2 Fig**).

proteasome expression [45–47], and the work of Geiler-Samerotte *et al.* showed that growth-predictive proteins are components of the cytosolic unfolded protein response [48], However, it is not clear if this is related to proliferation in-and-of-itself or due to aneuploidy and other

genetic alterations [49]. Our results suggest that coordinated regulation of the ribosome and proteasome is an intrinsic signature of fast proliferating cells that is conserved across cell-types and species.

To test if the coupling between ribosome biogenesis and proteasome expression holds across species and in diverse cell types, we analyzed the bulk RNA-seq data across developmental stages, covering multiple organs in seven species [17]. Ribosome biogenesis and proteasome expression are highly correlated (**Fig 3E**). The coordinated expression change with developmental stages between ribosome biogenesis genes and proteasome genes across organs and species suggests that the coordination between protein synthesis and degradation is likely to be a conserved feature across a large number of species and cell-types (**S2D Fig**).

In addition to ribosome-biogenesis and the proteasome, several other gene sets are differentially expressed between fast and slow proliferating cells in both fibroblasts and ESCs (**Fig 3C**). mTORC1 (mammalian Target Of Rapamycin Complex 1) functions as a nutrient sensor and regulator of protein synthesis, and is regulated by nutrient and cytokine conditions that cause differences in proliferation [50,51]. We find that, even in the absence of genetic and environmental differences, mTORC1 is more active in fast proliferating cells. Activation of mTORC1 can promote ribosome-biogenesis [50,52], however, there is still controversy about the regulation of proteasome activity by mTORC1 [51,53–57].

The transcription factor MYC (**S2C Fig**), and MYC target genes (**Fig 3C and S2 Table**) are more highly expressed in fast proliferating cells. MYC is frequently amplified in cancer, regulates the transcription of ~15% of all genes [58] and is a master regulator of cell proliferation [59]. Overexpression of MYC promotes ribosome-biogenesis and cell growth rates [60,61], and active mTORC1 can promote MYC activation [62,63]. Our data suggest that increased expression of MYC and increased mTORC1 activity are general properties of fast-proliferating cells, and those genetic or environmental perturbations are not necessary to cause differential expression of these pathways.

## Defining a proliferation signature to predict the growth rate across species

Expression of typical proliferation markers, such as PCNA and KI67, did not correlate with intra-population heterogeneity in proliferation (**Fig 4A**). The high degree of conservation of genes whose expression correlates with intra-proliferation heterogeneity in proliferation, from yeast to mouse, suggests that there should be a set of genes whose expression is predictive of growth rate across all eukaryotes. To build such a set we combined "proliferation correlated genes"–those with a Spearman correlation of rho = 1 in both fibroblasts and ESCs (243 genes) with genes from six ribosome biogenesis and proteasome related gene sets that are significantly enriched in both fibroblasts and ESCs, which result in a final gene set consisting of 370 genes (**S4 Table**). We then applied ssGSEA [64], a rank-based method that computes an overexpression measure for a gene set of interest relative to all other genes in the genome. The proliferation signature score is the Normalized Enrichment Score, calculated by ssGSEA, for the gene set containing all 370 proliferation-correlated genes.

To test the ability of this proliferation signature to predict proliferation rates in new data we used out-of-species cross-validation. While several models have been developed to predict growth rates from gene expression [30], the performance of these models has been evaluated using within-experiment cross validation, in which a single sample (e.g., condition or genotype) was held-out (excluded) and used for testing model performance. Accurate prediction of growth rates in cells for which actual growth rates cannot be measured, such as tumors *in-vivo*, or from single-cell RNA-sequencing data, would be more useful. However, models tested using in-experiment cross-validation (also known as internal validation) are often over-fit,

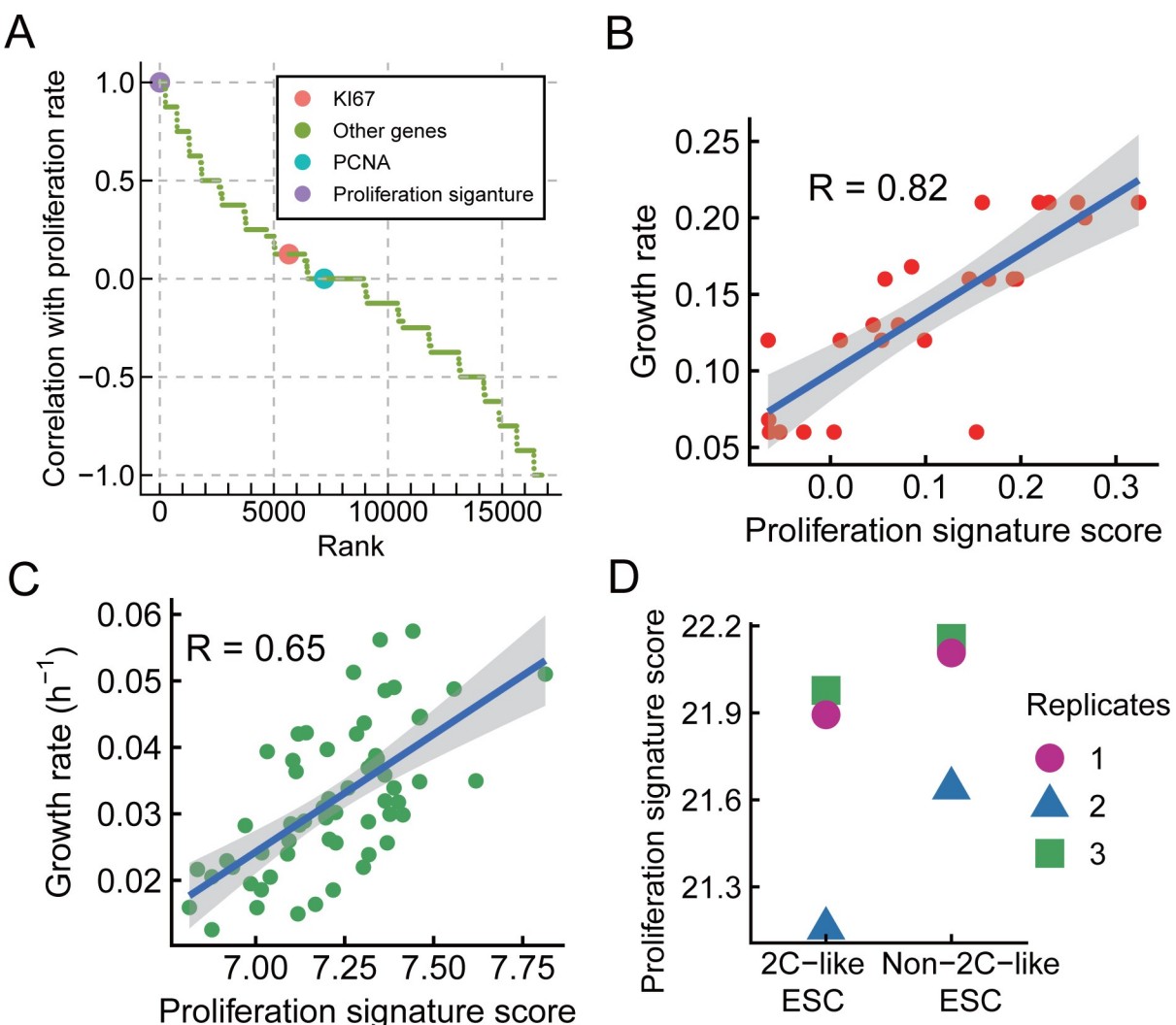

**Fig 4. A proliferation signature model can predict relative growth rates from gene expression for species and cell-types on which it was not trained. (A)** Genes and proliferation signature spearman correlation with proliferation rate (sorted by CFSE). Compare with KI67 or PCNA, proliferation signature has a better correlation with proliferation rate. **(B)** Using the proliferation signature to predict growth rate in budding yeast, we apply ssGSEA to calculate the enrichment score of proliferation signature for each sample. The Pearson correlation of proliferation signature score with growth rate is 0.82 (p = 8.9×10$^{-7}$), the grey shading is a 95% confidence band. **(C)** Using the proliferation signature to predict growth rate in cancer cell lines, the Pearson correlation is 0.65 (p = 1.9×10$^{-8}$), the grey shading is a 95% confidence band. **(D)** Comparison of proliferation signature score between 2C-like ESC and non-2C-like ESC (paired t-test, p = 0.04669).

resulting in poor performance when the model is applied to new data from new experiments [65].

To overcome this problem, we used the proliferation signature from above, which was developed using data from mouse cells, to predict growth rate from gene expression in budding yeast. Our model has correlations of R = 0.82, 0.73 and 0.77 across three different datasets (**Figs 4B, S3B and S3C**). In contrast, the model of Wytock et al. [30], which was trained on these yeast data, has an out-of experiment predictive power of R < 0.15 (Fig S6 in Wytock et al. [30]). The Wytock et al. model is over-fit and cannot predict proliferation rates in new data. Similarly, our model has better performance (R = 0.65) than a cancer-specific model [66], which was trained on cancer cell-line data and cannot predict out-of-experiment

(**Fig 4C**) (Fig 4 in Waldman et al. [66]). In contrast to most published models, our proliferation signature score model performs well on data on which it has not been trained.

When run on FACS-sorted 2C-like embryonic stem cells (2C::tdTomato+) [67], our proliferation signature model predicts that 2C-like mESCs proliferate slower (**Fig 4D**). 2-cell embryos also have uniquely low proliferation scores (**S4H Fig**). These results are experimentally independent of, and biologically consistent with, our observation that expression of 2C-like cell state marker genes is higher in slow proliferating mESCs (**Fig 2C**). This provides further evidence that the proliferation signature we have identified can be universally applied to predict the proliferative state of many cell types.

## Prediction of lineage-specific changes in proliferation rates during C. elegans development by the proliferation signature

Expression of the most commonly used markers (PCNA and KI67) for measuring proliferation rates in bulk populations are cell-cycle regulated, but the expression levels of these markers do not measure proliferation rates in single cells. Single-cell RNA sequencing is a powerful method for understanding development and differentiation *in vivo*, but it suffers from high levels of noise at the single-gene level [68]. We reasoned that our proliferation signature model would be ideal for measuring the proliferation rates of single cells from scRNA-sequencing data, as the model takes into account the expression of >300 genes, most of which are highly expressed and therefore have low levels of technical noise. To test the ability of the proliferation signature model on scRNAseq data we used a dataset of 89,701 cells from *C. elegans* development [69]. We computed the proliferation signature score for each cell and divided the cells into terminal cell types vs preterminal cells. Non-terminally differentiated cells have a higher proliferation signature score (t-test, p = $4.9 \times 10^{-41}$) (**Fig 5A**). A visual comparison of the 89,701 cells in UMAP space, colored by either embryo age [69] or by proliferation signature score (**Fig 5B**), suggested that, globally, proliferation rates in single cells decrease as development proceeds. However, three clusters of cells did not follow this pattern: germline, intestine and M cells. To quantify the relationship between proliferation rates of single cells and developmental time we binned all cells with same embryo time, and calculated the correlation between proliferation score and developmental time (**Fig 5C**). The proliferation signature score decreases as the embryo develops (rho = -0.65, p = $9.3 \times 10^{-19}$ and rho = -0.73, p = $8.7 \times 10^{-24}$ when excluding the three outlier groups (**S4B Fig**)). This conclusion from our single-cell gene expression analysis using the proliferation signature score is therefore quantitatively consistent with lineage-tracing microscopy data from Sulston et al. [70], showing that the rate of cell division within the developing nematode decreases during development (**S4D and S4E Fig**).

To our surprise, the predicted proliferation rates increased after 650 minutes (**S4C Fig**). To understand why, we grouped all cells in 650 minutes or older by lineage (**Fig 5D**). The three outlier groups from UMAP space: germline, M cell and intestine, had the highest proliferation signature among all cell types late in development. Specifically, for these three cell types, the proliferation score did not decrease with the embryo time, but increased or maintained a high level (**Fig 5E**). This can be explained by lineage-specific characteristics: the germline is the only cell type in *C. elegans* that is continuously proliferating, M cells are a highly proliferative single mesodermal blast cell, and the intestinal cells, although they do not proliferate, continue to increase in both biomass and DNA content through endoreplication. Other cell types with high proliferation scores, such as Z1/Z4, are also known to continue proliferation after 650 minutes [71]. The proliferation signature score decreases with embryo time for most cell types, including body wall cell, hypodermis and ciliated amphid neuron, which are the most prevalent in the single-cell RNA-seq dataset (**Fig 5E**).

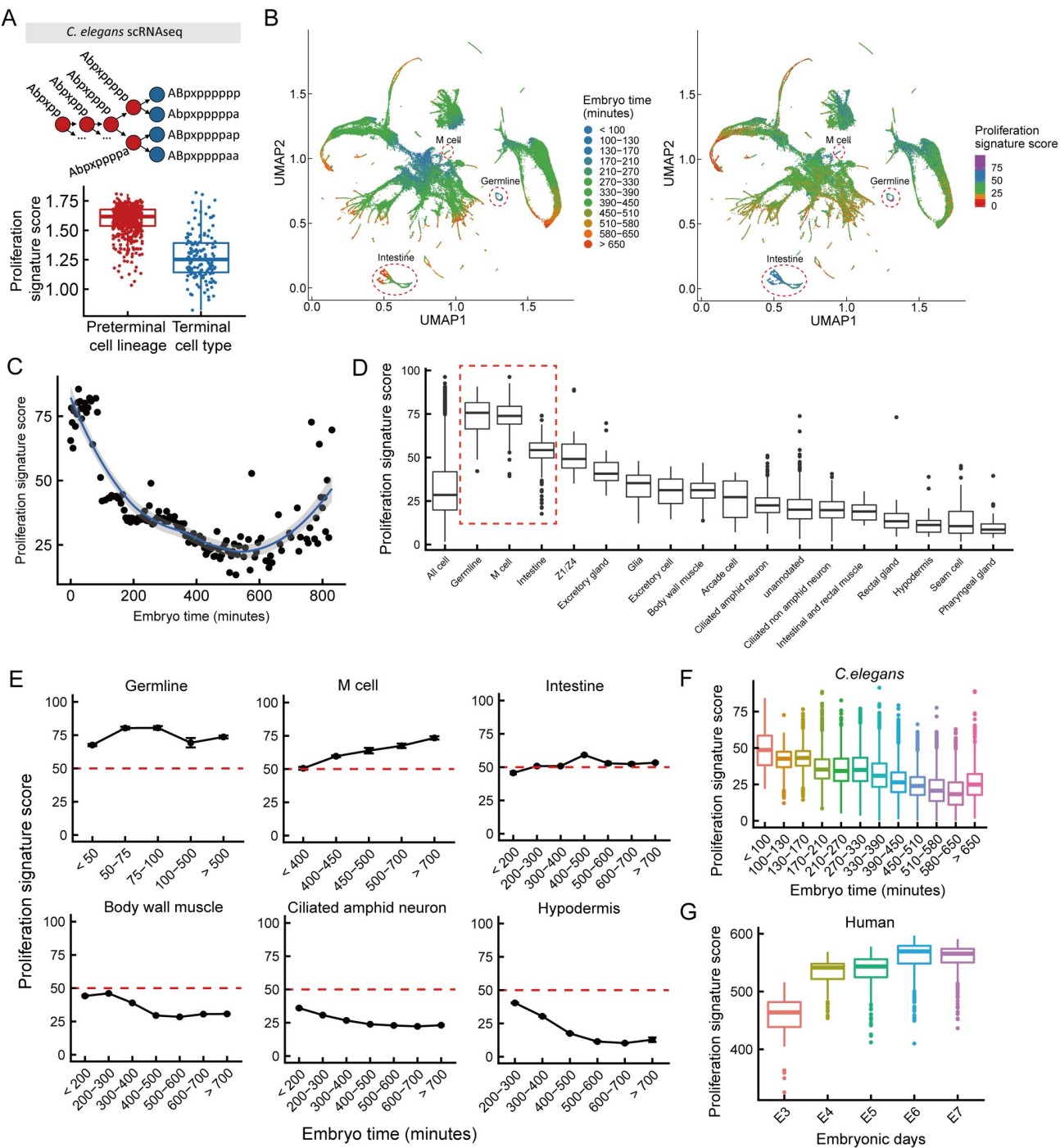

**Fig 5. Proliferation signature of cell development. (A)** A cartoon showing four terminal cells, and a partial linage showing the final four generations of preterminal cells. Comparison of single-cell proliferation signatures between preterminal cell lineage and terminal cell types (t-test, p = 4.9×10⁻⁴¹). **(B)** UMAP projection of 89,701 cells. Cells in the left panel are colored by estimated embryo times; in the right panel by proliferation signature score. **(C)** To calculate the proliferation signature score (y-axis) at each time point (x-axis) cells are binned by embryo time, and the mean proliferation signature score for all cells in the same bin is calculated. The spearman correlations are -0.65 (p = 9.3 ×10⁻¹⁹) for binned data and -0.42 (p < 2.2e-16) for unbinned data. **(D)** Boxplots (line shows median, boxes interquartile range) of proliferation signature score for all cells with embryo time > 650min. **(E)** Temporal dynamics of proliferation scores of select cell lineages, showing the average proliferation score for all single cells in that lineage, at each time point. **(F-G)** Boxplot of *C. elegans* **(F)** and human **(G)** proliferation signatures as a function of developmental time, from scRNAseq data.

Two cell types, hypodermis and seam cells, exhibited very low proliferation signature scores at late time points, while intestinal cells exhibited very high proliferation scores (**Figs 5E and S4F**). Both these cell types contain multinucleated cells, but these multinucleated cells arise through very different mechanisms: hypodermis and seam cells through cell-fusion, and intestine through endoreplication [71]. Thus, two cell types with seemingly similar properties have highly divergent transcriptomes, and highly divergent mechanisms to reach their similar final state.

Upon development to an L1 larva, a nematode has more than half of the final number of cells present in the adult. In contrast, mammals continue to rapidly increase in cell number even after embryonic development is complete. This difference can be seen in the change in proliferation signature over time, which decreases in *C. elegans*, but increases in human (data from Petropoulos et al. [72]) and mouse (data from Deng et al. [73]) (**Figs 5F–5G and S4G and S4H**). In conclusion, our proliferation signature genes obtained from mouse fibroblast and ESC data can predict dynamic changes in proliferation rates during the development of various cell types and species, thereby confirming its universal applicability.

## Cell-to-cell heterogeneity in mitochondria state predicts variation in proliferation both in ESCs and fibroblasts, but in opposite directions

While the pattern of within-population proliferation-correlated expression in yeast, mouse fibroblasts and ESCs was broadly similar with regard to genes involved in protein synthesis and degradation, the behavior of metabolic and mitochondria-related genes in fast and slow proliferating subpopulations was highly cell-type specific. Mitochondria membrane and respiratory chain-related gene sets were more highly expressed in fast proliferating fibroblasts, but not in fast proliferating ESCs (**Table 1**). These results are consistent with differential mitochondrial states in ESCs when compared to differentiated cells like fibroblasts [23], which suggests the existence of different types of metabolism and proliferation-related heterogeneity between pluripotent and differentiated cell-types. We also observed cell-type specific differences in glycolysis, fatty acid metabolism, and other metabolic processes, suggesting fundamental differences in the metabolic pathways required for fast proliferation between pluripotent ESCs and differentiated cells like fibroblasts (**Table 1**).

The mitochondrial membrane potential is a major predictor of cell-to-cell heterogeneity in proliferation rate in budding yeast [9]. Mitochondria-related genes are more highly expressed in the fast proliferating subpopulation of fibroblasts (**Table 1**). In contrast, these genes are slightly more highly expressed in the slow proliferating subpopulation of ESCs. This suggests that the relation between cell-to-cell heterogeneity in mitochondria state and proliferation may be different in these two cell types. To test the ability of mitochondrial membrane potential to predict proliferation rate in mammalian cells we used the mitochondria membrane potential-specific dye tetramethylrhodamine ethyl ester (TMRE) to stain fibroblasts and ESCs, and performed both RNA-seq and proliferation-rate assays on high and low TMRE sub-populations (Fig 6A).

Unlike the proliferation-based sort (**Fig 1**), sorting ESCs and fibroblasts by mitochondria-state (**Figs 6 and S5A and S5B**) resulted in highly divergent expression profiles. ESCs with high TMRE signal had high expression of ribosome-biogenesis, proteasome, MYC-targets and mitochondrial-related genes, while in fibroblasts these gene sets are more highly expressed in the low TMRE sub-population (**Fig 6B and 6C and S5 Table**). This is consistent with the opposite behavior of mitochondria-related gene sets in proliferation-rate sorted cells from the two cell types (**Table 1**). We note that the differences between high and low TMRE populations are smaller than the difference between high and low CFSE populations (**S5C and S5D Fig**),

**Table 1. Gene sets whose expression exhibits opposite correlations with growth between fibroblasts and ESCs.**

| | | | Fibroblasts | | ESCs | |
|---|---|---|---|---|---|---|
| | Gene set name | Gene set size | NES | FDR.q.val | NES | FDR.q.val |
| **Mitochondria** | Inner mitochondrial membrane protein complex | 101 | 2,52 | <0.001 | -0,39 | >0.1 |
| | Mitochondrial membrane part | 164 | 2,26 | <0.001 | -0,44 | >0.1 |
| | Mitochondrial respiratory chain complex assembly | 74 | 2,19 | <0.001 | -0,49 | >0.1 |
| | Mitochondrial respiratory chain complex I biogenesis | 54 | 2,12 | <0.001 | -0,49 | >0.1 |
| | Mitochondrial matrix | 404 | 1,97 | <0.05 | -0,49 | >0.1 |
| **Metabolism** | Metabolism of proteins | 377 | 2,47 | <0.001 | -0,54 | >0.1 |
| | Glycolysis gluconeogenesis | 60 | 2,03 | <0.05 | -1,35 | >0.1 |
| | Monosaccharide biosynthetic process | 52 | 1,94 | <0.05 | -0,76 | >0.1 |
| | Monosaccharide catabolic process | 56 | 1,67 | <0.05 | -0,93 | >0.1 |
| | Hallmark fatty acid metabolism | 157 | 1,52 | <0.1 | -0,71 | >0.1 |
| **Differentiation** | Dopaminergic neuron differentiation | 28 | -1,66 | <0.05 | 1,27 | >0.1 |
| | Hematopoietic progenitor cell differentiation | 97 | -1,59 | <0.1 | 1,09 | >0.1 |
| | Regulation of cardiac muscle cell differentiation | 19 | -1,57 | <0.1 | 0,92 | >0.1 |
| | regulation of smooth muscle cell differentiation | 20 | -1,45 | >0.1 | 1,66 | <0.1 |
| | Glial cell differentiation | 136 | -1,00 | >0.1 | 1,63 | <0.1 |
| **Cell cycle** | Cell cycle G1 S phase transition | 104 | -1,95 | <0.05 | 2,03 | <0.05 |
| | Hallmark E2F targets | 195 | -2,09 | <0.001 | 2,43 | <0.001 |
| | Fischer G1 S cell cycle | 177 | -2,03 | <0.001 | 1,90 | <0.05 |
| | Cell cycle checkpoints | 110 | -0,84 | >0.1 | 1,82 | <0.05 |
| | Cell cycle phase transition | 247 | -1,99 | <0.001 | 1,31 | >0.1 |

Shown are representative gene sets whose expression is significantly correlated with proliferation in either fibroblasts or ESCs, but whose expression changes with proliferation in opposing directions. NES>0 (higher expression in fast); NES<0 (higher expression in slow)

either due to technical limitation of the dye, or because there is less heterogeneity in mitochondria state than there is in proliferation rate.

These expression data make the following prediction: ESCs with high TMRE should have a shorter doubling time, while fibroblasts with high TMRE should have a longer doubling time. To test this, we sorted fibroblasts and ESCs by TMRE, and found that high TMRE fibroblasts indeed do proliferate more slowly, while high TMRE ESCs proliferate more rapidly (**Fig 6D**). In addition, we tested the effect of ascorbic acid (vitamin C, an antioxidant) and $O_2$ levels (ambient 21% atmospheric vs. low 5% physiological levels) on doubling time, but found no significant effects (**S6 Table**).

To investigate additional cell types, we searched for RNA-seq data for cells sorted by mitochondria state, and analyzed RNA-seq data of mouse CD8+ T-lymphocytes that have been sorted by mitochondria membrane potential (TMRM) [22]. CD8+ T cells with high TMRM signal (high $\Delta\Psi$m) showed higher expression of ribosome-biogenesis and proteasome-related genes (**S7 Table**), and proliferate more rapidly [22], thereby behaving similarly to ESCs.

Thus, across yeast, mouse ESCs, fibroblasts and CD8+ T cells, while mitochondria state and proliferation rate co-vary within a single population, the direction of this correlation is different, with yeast and fibroblasts behaving similarly with each other, and opposite to ESCs and CD8+ T cells.

## Perturbation of mitochondria function affects fast and slow proliferating fibroblasts and ESCs in different ways

To investigate the relationship between proliferation rate, cell type, and mitochondrial state, we performed perturbation experiments by directly inhibiting mitochondria function. We

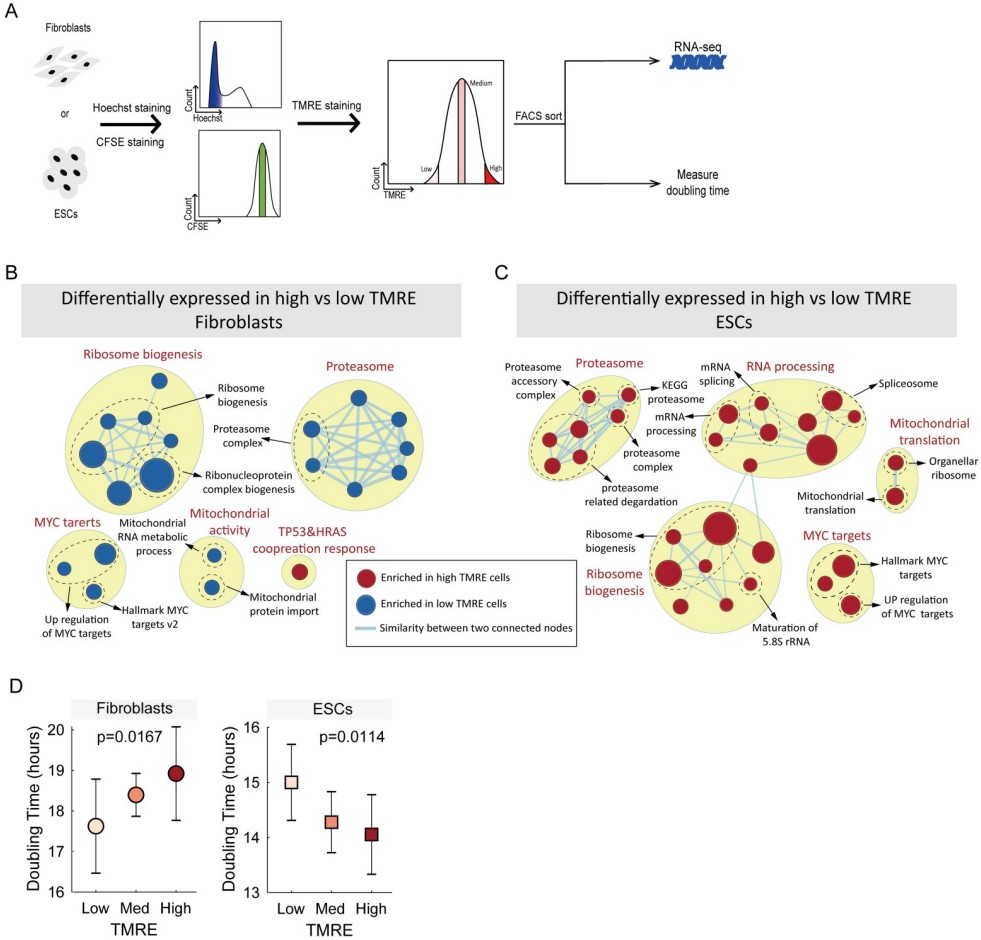

**Fig 6. Expression of proliferation-related gene sets in cells sorted by intra-population heterogeneity in mitochondria membrane potential. (A)** Cells were stained with Hoechst and CFSE and a homogenous population of equally sized cells in G1 with equal CFSE was obtained by FACS. These cells were stained with TMRE sorted by TMRE, and then used for RNA-seq, or allowed to proliferate to measure the doubling time of each TMRE sub-population. **(B, C)** Enrichment maps of fibroblasts and ESCs sorted by TMRE. **(D)** Doublings times, as estimated by the measured by the decrease in CFSE signal over time, for high, medium and low TMRE sorted cells. P-values are from ANOVA, testing if TMRE is predictive of doubling time (see Materials and Methods).

stained both mouse ESCs and fibroblasts with CFSE and sorted 20% of the viable cells on the peak of the CFSE signal to have a homogeneous starting population. After culturing them for 24h or 48h respectively, two bins were sorted: the lowest 20% (fast proliferating cells) and the highest 20% CFSE (slow proliferating cells) (**Fig 7A**). Next we cultured the sorted cells for 8h, then treated with either medium containing DMSO as mock control, the mitochondrial electron transport chain complex III inhibitor antimycin, the ATP synthase inhibitor oligomycin for 16h, washed out the drugs, and measured both viability and proliferation rate for every 24h.

Both fast fibroblasts and ESCs sorted by CFSE signal maintained a higher fraction of cells in S-phase over two days in growth-media with DMSO, indicating that the proliferation status, fast vs slow, is semi-heritable (**Fig 7B**). Interestingly, we found cell-type and proliferation-state specific effects of mitochondria perturbation. Antimycin A, an inhibitor of complex III of the electron transport chain, strongly decreased the fraction of slow-proliferating fibroblasts that were in the S-phase but had a weaker effect on fast-proliferating fibroblasts (t-test, p = 0.0089)

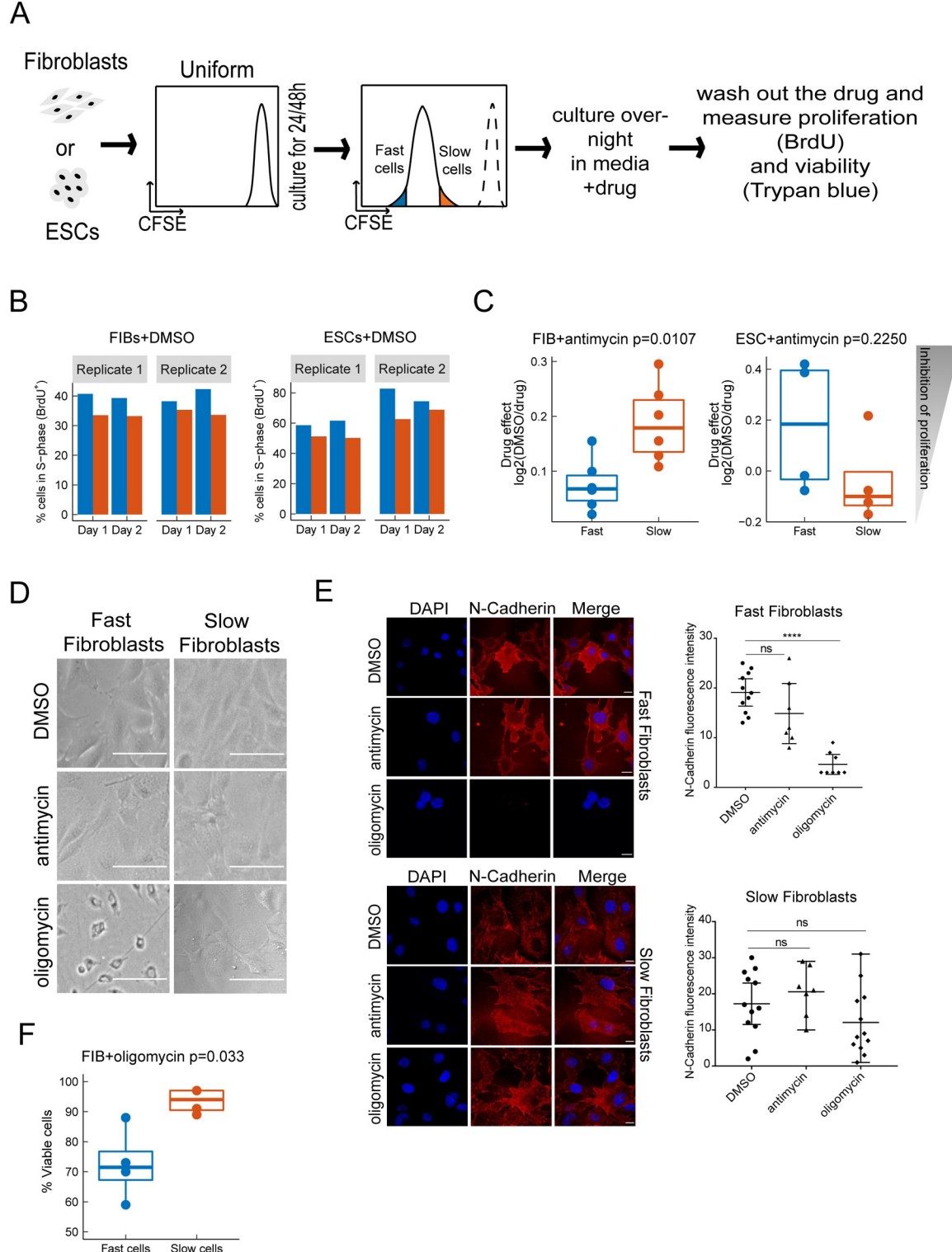

**Fig 7. Growth-rate and cell-type specific effects of mitochondria inhibitors on proliferation rate, cell viability and cell state. (A)** Schematic of the experimental setup for measuring the effects of mitochondria inhibitors on slow and fast proliferating cells. **(B)** Fast proliferating Fibroblast and ESCs sorted by CFSE signal maintained a higher fraction of cells in S phase over two days of growth in medium+DMSO. **(C)** Effect of antimycin treatment on fast and slow proliferating fibroblasts and ESCs (Drug effect: log2 of the fraction of cells in S-phase when treated with DMSO divided by the fraction of cells in S-phase when treated with drug). **(D)** Fast fibroblasts changed morphology after the treatment with oligomycin. Scale bars = 80 μm. **(E)** Immunostaining of fibroblasts for N-Cadherin and

DAPI after drug treatment and corresponding quantifications. Fast fibroblasts lose N-Cadherin staining specifically after oligomycin treatment. Scale bars = 15 μm. The fluorescence intensity of N-Cadherin has been quantified on the right. Medians and the 95% confidence intervals are shown as error bars. Kruskal–Wallis test was used for statistical comparison (ns, not significant, **** p < 0.0001). **(F)** Effect of oligomycin treatment on fibroblast viability. The % viable cells is measured as % trypan blue-negative cells.

**(Fig 7C)**. In ESCs, the effect of antimycin appeared somewhat stronger on fast- than on slow-proliferating cells (although it did not differ significantly).

In contrast, fast proliferating fibroblasts were highly sensitive to oligomycin treatment, which blocks the ATP-synthase of complex V in the electron transport chain. Specifically, we observed a change in cell morphology upon treatment **(Fig 7D)**. In comparison with DMSO-treated cells, the cells lost their elongated shape and became more round and smaller. This led us to hypothesize if that morphology change might be explained by a mesenchymal to epithelial transition (MET) upon oligomycin treatment. In fact, during induced pluripotent stem cell (iPSC) reprogramming, MET of fibroblasts is an important early reprogramming step [74,75]. In that context, oligomycin treatment has been recently shown not only to promote a metabolic switch from oxidative phosphorylation to glycolysis, but also to modulate mesenchymal markers during reprogramming [75,76]. Therefore we measured the protein levels of the regulators N-cadherin (mesenchymal marker expressed in fibroblasts) and E-cadherin (epithelial marker expressed in ESCs) with and without treatment, by both immunostaining and Western Blot **(Figs 7E, S6A and S6B)**. We could not detect E-cadherin in fibroblasts, but we observed reduced expression of N-cadherin in comparison with DMSO treated-cells in particular in oligomycin-treated fast cycling fibroblasts **(Figs 7E, S6A and S6B)**. In addition to the change in morphology, oligomycin treatment reduced cell viability specifically in fast proliferating fibroblasts, but not in slow fibroblasts **(Fig 7F)**. In conclusion, although we observed both a change in cell viability, morphology and a reduction in N-cadherin levels, oligomycin treatment did not induce a complete mesenchymal to epithelial transition in the fast-proliferating fibroblasts as indicated by the lack of E-cadherin upregulation. Antimycin and oligomycin interfere with different parts of the electron transport chain of distinct functions—complex III, oxidation of substrates and setting up membrane potential vs. complex V, reduction of membrane potential and ATP-production. Therefore, the distinct effects of antimycin treatment on the proliferation of fibroblasts and ESCs and the subpopulation-specific effect of oligomycin on fast fibroblasts are in line with a differential dependency on the diverse mitochondrial functions between the different subpopulations of fibroblasts and ESCs.

## Discussion

In summary, we have developed a method to sort cells by their proliferation rate and used these data to identify a pattern of proliferation-correlated gene expression that is conserved among eukaryotes. We used these data to develop a model that can predict proliferation rates from gene expression in multiple eukaryotic species and cell types, and for types of data, such as single-cell RNA sequencing in a developing organism, for which proliferation rates cannot be measured experimentally.

While the CFSE signal is not a measure of the instantaneous proliferation rate, but instead determined by the average proliferation rate integrated over several days, the fact that (A) the transcriptomes of the sorted cells are predictive of proliferation rates, and (B) the cells with low CFSE maintain faster proliferation rates over at least three days, suggests that there are not likely to be large differences in the instantaneous proliferation rate vs the average rate, at least for these cell types and experimental timescales.

We found that genes involved in protein synthesis (ribosome-biogenesis, translation initiation), and in protein degradation (the proteasome and proteasome-related protein degradation) are highly expressed in fast proliferating eukaryotic cells, including mammalian, nematode and yeast cells. Previous studies have reported that high expression of the proteasome in fast-growing cells may be necessary in order to degrade misfolded protein, because the fast protein synthesis in fast-growing cells produces more incorrectly folded proteins [51,77,78]. Even with a constant translation and folding error rate, fast proliferating cells will produce more protein, and therefore more misfolded protein that needs to be degraded.

In all non-cancer mammalian cells we investigated, we also found the mTORC1 signaling pathway enriched in fast proliferating cells and P53-targets enriched in slow proliferating cells. Our results show both upregulations of the mTORC1 signaling pathway and proteasome activity in fast proliferating cells, which is consistent with several previous studies [9,12–15].

Our analysis of fast versus slow proliferating ESCs cultured in 2i+LIF conditions indicated at several levels that slow proliferating cells were of a more naïve ground state pluripotent character than fast proliferating cells. First, this was supported by the fact that they displayed a higher expression of naïve pluripotency marker genes and markers of 2C-like cells (**Fig 2B and 2C**). Second, we observed enrichment of E2F targets and genes involved in G1 S cell cycle phase transition (**Table 1**) in our fast cycling ESC population, indicative of a shortened G1 phase and a shorter doubling-time, as described for ESCs cultured in serum+LIF [27]. Finally, although we could find differentiation genes to be expressed both in fast and slow proliferating cells (**Fig 2A**), we saw several differentiation pathways to be enriched specifically in fast dividing ESCs (**Table 1**). In summary, even when ESCs are cultured in ground-state pluripotency promoting 2i+LIF conditions, they display heterogeneity in proliferation rate, with the slow proliferating being of more naïve pluripotent character when compared to fast dividing cells.

While we observed ESCs behave similarly to other cell types like fibroblasts or yeast when it comes to gene expression signatures characteristic of fast proliferating cells related to protein synthesis and turnover (**Fig 3C**), we found a very different behavior when it comes to regulation of metabolism. Although the growth rate can be predicted by mitochondrial membrane potential in *Saccharomyces cerevisiae* [23], where it is negatively correlated with proliferation rate like in fibroblasts as we show in this study, our results show mitochondrial membrane potential to be positively correlated with proliferation rate in ESCs (**Fig 6D**). This suggests mitochondrial membrane potential has different functions in pluripotent cells when compared to differentiated cell types or yeast. This is corroborated by our gene expression analysis of cells with high vs. low mitochondrial membrane potential (**Fig 6B and 6C**), where we found pathways linked with fast proliferating cells to be enriched in fibroblasts with low mitochondrial membrane potential but on the contrary, enriched in ESCs with high mitochondrial membrane potential. Surprisingly, primed pluripotent stem cells have been described to rely more on non-oxidative, glycolysis-based metabolism than naïve pluripotent stem cells [79–81], which appears in contradiction with our result that our slow proliferating, mitochondria activity low ESCs being more naïve-like. However, TMRE is not a direct measure of ATP generation by mitochondria; yeast cells that are respiring and producing all of their ATP using their mitochondria, and yeast cells unable to respire, both have high TMRE signals [9]. Differentiated cells, in general, rely more on oxidative metabolism than pluripotent cells, therefore our fast proliferating ESCs could potentially reflect a more differentiation prone state.

*In vivo*, cells exhibit a great degree of variability in proliferation rates, from terminally differentiated neurons, to slowly proliferating cancer stem cells, to rapidly proliferating embryonic stem cells. Many cell types, such as hematopoietic stem cells, contain both proliferating and non-proliferating populations. The proliferation signature model, because of its applicability across all tested species and cell types, provides a useful tool for understanding *in vivo*

development for systems, in which precise measurements of proliferation are impossible. Our model has been validated on scRNA-seq data, using published time-lapse microscopy of cell lineages in *C. elegans* as the ground truth [70]. However, it is technically challenging to do microscopy or to otherwise measure proliferation of individual cells inside of a developing mouse embryo, or in a tumor in a patient. Models that can accurately predict difficult to measure properties, such as proliferation rate, from easy to measure ones, such as gene expression, will therefore aid in our understanding of complex biological processing during tumor formation, differentiation, and development.

## Materials and methods

### Cell culture growth conditions

Tail tip fibroblasts (TTFs) were isolated from a female newborn mouse from a *Mus musculus x Mus Castaneus* cross and immortalized with SV40 large T antigen [82]. The clonal line 68-5-11 [83] was established and maintained in DMEM supplemented with 10% serum (LifeTech), HEPES (30mM, Life Tech), Sodium Pyruvate (1mM, Life Tech), non-essential amino acids (NEAA) (Life Tech), penicillin-streptomycin (Ibian Tech), 2-mercaptoethanol (0.1mM, Life Tech).

The mouse embryonic stem cell (ESC) line EL16.7 (40XX, Mus musculus/*M.castaneus* hybrid background [84] was maintained on gelatin coated tissue culture dishes and passaged every 2 days by seeding around $2x10^6$ cells in 2i+LIF medium. Accutase (Merck Chemicals and Life Science) was regularly used for cell detachment when passaging cells. 2i+LIF medium contains a 1:1 mixture of DMEM/F12 supplemented with N2 (LifeTech) and neurobasal media (LifeTech) supplemented with glutamine (LifeTech), B27 (LifeTech), insulin (Sigma), penicillin-streptomycin (Ibian Tech), 2-mercaptoethanol (LifeTech), based on previous reports [37]. It also has additional modifications reported in Hayashi et al. [85] containing two chemical inhibitors 0.4 μM PD032591 (Selleck Chemicals, S1036) and 3 μM CHIR99021 (SML1046, SML1046) together with 1,000 U/ml LIF (ORF Genetics, 01-A1140-0100). Both TTFs and EL16.7 were cultured at 37˚C in 5% $CO_2$.

### Proliferation and doubling time analysis

ESCs and fibroblasts were plated on 10 cm plates at $5.3x10^6$ and $7.3x10^5$ concentrations, respectively. Cells were expanded and counted for 7 days. To monitor distinct generations of proliferating cells, carboxyfluorescein succinimidyl ester (CFSE, Thermo Fisher Scientific) was used to stain the cells and the dilution of the dye was detected by flow cytometry every day. CFSE was dissolved in dimethyl sulfoxide at a concentration of 5 mM as stock solution and CFSE was added to a 1 ml cell suspension, to a final concentration of 5uM or 10uM. After the addition of CFSE, cells were incubated at 37˚C for 20 min. Then the cells were washed twice with complete medium and maintained on ice until use in a buffer containing PBS, 2% serum and 1% pen-strep. Cell viability was determined by DAPI (Biogen Cientifica) staining. Dye signals were measured on an LSRII flow cytometer.

### RNA-seq

To collect cells with different growth rates, cells were isolated by sorting at room temperature according to the CFSE signal (median and high CFSE signal). ESCs and fibroblasts were sorted into 1.5ml Eppendorf tubes containing medium and were cultured for 3 days and 5 days respectively in specific culture conditions as described earlier. For each cell line three bins were sorted: the lowest 10%, the median 10% and the highest 10% CFSE. Cells were sorted into

prechilled 1.5ml Eppendorf tubes containing 200μl medium each. Cells were then centrifuged at 1000 rpm for 5 min, the media removed and the resulting cell pellet was used for RNA extraction. All bins were treated identically throughout the process. Cellular RNA was extracted using the Maxwell RNA Purification Kit and processed for RNA sequencing. For biological replicates, all experiments were repeated on three or four different days. Expression was quantified using Kallisto v0.42.3 [86] from the raw reads (no pre-processing) using the gencode.VM18.transcript annotations. We experimented with multiple methods for batch effect removal using the R package SVA [87] and found that the results of the GSEA, with regards to which gene sets were differentially expressed between fast and slow, or high and low TMRE cell populations, did not change. We therefore used the original data.

PCA on TMRE sorted biological replicates showed that one TMRE sort was an extreme outlier (S5E Fig); this pair was discarded from all analysis.

## BrdU Staining

Cell Proliferation was measured by the incorporation of bromodeoxyuridine (BrdU). Every 24h BrdU was added at a final concentration of 10 μM to the cells. Incubation under the appropriate growth conditions occurred for 30 minutes for ESCs and 45 minutes for fibroblasts to pulse label the cells. Cells were trypsinized, spun down at 1050 rpm for 5 minutes. After washing them in ice-cold PBS, cells were fixed overnight in ice cold Ethanol (70%) while maintaining a gentle vortex. The following day the Ethanol fixed cells were centrifuged and the DNA denatured by adding 2N HCl—0.5% Triton X-100 for 30min at room temperature. Then cells were centrifuged and resuspended in 0.1 M $Na_2B_4O_7$, pH 8.5 for 10 minutes at room temperature. After spinning them down the cells were incubated overnight at 4˚C with PerCP/Cy5.5 anti-BrdU (1:30 dilution) (BioLegend) in a buffer containing 0.5% Tween 20 / 1% BSA/PBS and RNase (0.8 mg/ml). The following day cells were washed in PBS and spun down at 1050rpm for 5min at room temperature. The pellet was resuspended in PBS with DAPI (1:1000) and analyzed in an BD LSRII flow cytometer.

## Mitochondria inhibitor treatment assay

For the assessment of chemical inhibitors on membrane potential changes, cells were incubated with medium containing DMSO (0.1%, mock control), antimycin A (500nM), oligomycin (1 μM) for 16h. Cells with or without treatment were washed with PBS and trypsinized. After spinning the cells for 5 minutes at 1050 rpm at room temperature, the cell pellet has been stained with 50nM TMRE for 20 min at 37˚C. After 2 times washes with PBS, cells were resuspended in PBS containing DAPI and immediately analyzed by flow cytometer BD LSRII.

In order to measure cell viability the trypan blue exclusion assay has been used. Upon trypsinization, an aliquot of the cell suspension has been mixed at 1:1 with trypan blue solution (0.4%, Thermo Fisher Scientific). The mix has been loaded on the counting chamber slide and analyzed at the automated counter (Invitrogen Countess II FL Automated Cell Counter).

## Western blot

Cells were lysed in a lysis buffer (20 mM Tris-HCL, pH 8.0, 150 mM NaCl, 1% Triton X-100, supplemented with protease inhibitors cocktail) and centrifuged for 30 minutes at 16000g. The supernatant was boiled in SDS loading buffer. After SDS-PAGE, the samples were transferred to a polyvinylidene difluoride membrane using a transfer apparatus according to the manufacturer's protocols (Bio-Rad). After incubation with 5% nonfat milk in TBST (10 mM Tris, pH 8.0, 150 mM NaCl, 0.5% Tween 20) for 1h, the membrane was washed once with TBST and incubated with antibodies against N-Cadherin (BD Biosciences, 1:1000), E-Cadherin (BD

Biosciences, 1:1000), GAPDH (Santa Cruz, 1:5000), at 4°C for 16 h. Membranes were washed three times for 10 min and incubated with a 1:5000 dilution of Rabbit Anti-Mouse Immunoglobulins/HRP for 1.5 h. Blots were washed with TBST three times and developed with the ECL system (Amersham Biosciences) according to the manufacturer's protocols.

## Immunofluorescence staining

Cells were grown in 8-well Lab-Tek chamber slides (Thermo Fisher Scientific) and fixed in 4% paraformaldehyde for 10min at room temperature. Then, washed three times in PBS. Fixed cells were permeabilized in 0.5% Triton X-100 (Sigma-Aldrich) in PBS buffer for 10min at room temperature. And then washed in PBST (PBS with 0.1% Tween (Sigma-Aldrich)) for 2min at RT. Then cells were incubated in a blocking solution containing 10% bovine serum albumin (BSA, Sigma) and 0.01% Triton X-100 for 1h at room temperature. Cells were then left at 4°C overnight in a blocking solution containing the primary antibody: mouse E-Cadherin (BD Biosciences, 1:1000) and mouse N-Cadherin (BD Biosciences, 1:1000). The next day, the cells were washed three times in PBS and then incubated with the secondary antibody for 45min at room temperature. Goat anti-mouse IgG, (1:1000, Life Technologies) conjugated to Alexa Fluor-555, was used as a secondary antibody. Nuclear staining was performed with DAPI (1:1000, Biogen Cientifica). Images were taken with a Leica TCS SP8 confocal microscopy system and were analyzed with Fiji (ImageJ).

## Differential expression of pluripotency, 2C-like state and lineage commitment-related genes in mESCs sorted by proliferation rate (CFSE)

Pluripotent state gene markers are chosen from 4 different studies [26, 29, 88, 89], only genes that are used as pluripotent state gene marker in at least 2 of these 4 papers are used in this paper. Lineage commitment and 2C-like state gene markers are the same as genes in Fig 5A of Kolodziejczyk et al. [26] and key differentiation regulators in **Fig 6** of the same study. To see the corresponding pluripotent cell state of fast and slow proliferating mESCs, we calculated the mean expression of naïve pluripotent markers in four fast-proliferating and four slow-proliferating replicates and log2(fast/slow) was calculated to compare genes expression in fast proliferating subpopulation and slow proliferating sub-population. The same method was applied to lineage commitment gene markers and 2C-like state gene markers.

## Mitochondrial membrane potential measurements

The relative mitochondrial transmembrane potential (ΔΨm) was measured using the membrane-potential-dependent fluorescent dye TMRE (Tetramethylrhodamine, Ethyl Ester, Perchlorate) (Molecular Probes, Thermo Fisher Scientific) [90]. For TMRE staining fibroblasts and ESCs were grown, washed in PBS, trypsinized and resuspended in PBS with 0.1% BSA and TMRE added at a final concentration of 50nM, from a 10μM stock dissolved in DMSO. Cells were incubated for 20min at 37°C, washed with PBS and were analyzed by flow cytometry or sorted.

## Cell sorting

Cells at 80% confluence in 10cm plates were trypsinized, centrifuged at 1000rpm for 5min and stained with medium containing 10μM CFSE for 20min. Then cells were washed twice with PBS and stained with DAPI as viability dye. To have a homogeneous starting population, 20% of the viable cells were sorted according to the proliferation rate on the peak of CFSE signal and re-plated. ESCs and fibroblasts have been cultured for 24h or 48h respectively and two bins were sorted: the lowest 20% (fast proliferating cells) and the highest 20% CFSE (slow

proliferating cells). Cells were sorted into prechilled 1.5-ml Eppendorf tubes containing 200 μl medium each. Cells were then centrifuged at 1000 rpm for 5 min, the media removed and plated in their culture medium. To monitor their proliferation rate, the dilution of the CFSE dye was detected by flow cytometry every day up to 3 days for ESCs and 5 days for fibroblasts. Dye signals were measured on an LSRII flow cytometer.

For the CFSE sort (no TMRE), cells were stained with CFSE and DAPI, and we used FACS to obtain a population of viable cells with the same CFSE signal. We then grew cells for 3 or 5 days, and every 24 hours measured the CFSE signal using flow cytometry. Staining did not have a strong effect on cell viability or proliferation (**S7 Fig**).

For the TMRE sort for proliferation rate, cells were stained with CFSE and TO-PRO-3, and we used FACS to obtain a population of G1 cells with the same CFSE signal. We then grow cells for 3 or 5 days, and every 24 hours measured the CFSE signal using flow cytometry.

In order to have a homogeneous starting population, both cell types were stained with Hoechst (10 ug/ml, Life Technologies) to pick cells in G0/G1 phase. Within this population, cells were selected according to the proliferation rate on the peak of CFSE signal prior to staining them with the dye. Then cells were sorted by TMRE into three bins: low, medium and high with a BD Influx cell sorter into prechilled 1.5 ml Eppendorf tubes containing 200 μl medium each. Cells were then centrifuged at 1000 rpm for 5 min, the cell pellet was washed with PBS and used for RNA extraction. All bins were treated identically throughout the process. Cellular RNA was extracted using the Maxwell RNA Purification Kit and processed for RNA sequencing. Cell viability was determined by TO-PRO-3 (Thermo Fisher Scientific) staining.

To test the effect of $O_2$ levels and ascorbic acid/vitamin C in both cell types, sorted cells from each bin were plated into each of the four different conditions (low $O_2$ (5%), normal oxygen growing conditions, and with or without ascorbic acid/vitamin C (25 ug/ml, Sigma-Aldrich)) in duplicate. After one day of recovery from the sorting, the cells were washed in PBS, were trypsinized, and counted. After seeding the same initial number, the rest of the cells was analyzed on a BD Fortessa analyzer. Every day a sample from each condition and replicate was taken for counting, and stained with 50 nM TMRE, up to 3 days for ESCs and 5 days for fibroblasts, and both TMRE and CFSE were measured by flow cytometry.

Images of CFSE and TMRE stained cells are shown in **S8 and S9 Figs**.

## Gene set enrichment analysis (GSEA)

GSEA was performed using the GSEA software and the MSigDB (Molecular Signature Database v6.2) [41,91]. We use signal-to-noise (requires at least three replicates) or log2 ratio of classes (for experiments with less than three replicates) to calculate the rank of each gene. The maximum number of genes in each gene set size was set to 500, the minimum to 15, and GSEA was run with 1000 permutations. We provided all GSEA results in this study (**S8 Table**).

## Enrichment map

Enrichment maps of this study (**Figs 3C, 3D, 6B and 6C**) are created using EnrichmentMap in Cytoscape [92,93], we refer to Reimand et al's protocol [94] for using EnrichmentMap.

We imported the output file of GSEA to EnrichmentMap and set FDR threshold as 0.1, other parameters set as default.

## Coordination of expression of ribosome biogenesis and proteasome related genes

We first calculate the mean expression (average of log2(TPM+1)) of ribosome biogenesis genes (genes in GO preribosome gene set) and proteasome genes (genes in GO proteasome

complex gene set) across organ developmental time course, then we calculate the Pearson correlation of ribosome biogenesis and proteasome.

## Calculation of proliferation signature scores

To obtain proliferation correlated genes, we first calculate, for each gene, the Spearman correlation with proliferation rate for each replicate, as measured by the decrease in CFSE signal, in both fibroblasts and ESCs. We define "proliferation correlated genes" as genes that have a correlation of 1 in all replicates (two replicates for both fibroblasts and ESCs, totally four sets of samples) and in both fibroblasts and ESCs (243 genes). The formula to decide the genes to be chosen as "proliferation correlated genes" is:

$$\mathrm{mean}(\mathrm{Cor}(gene(i), pr(i))) = 1$$

where $gene(i)$ is the gene expression for all samples with different proliferation rates (slow, medium and fast) in replicate $i$, $pr(i)$ is the proliferation rate for all samples in replicate $i$. As we have two replicates for each of the two cell types, which means we have four replicates, and the mean of correlation equal to 1 means the correlations in each replicate is rho = 1.

To this set we add genes from six ribosome biogenesis and proteasome related gene sets that are significantly enriched in both fibroblasts and ESCs, which result in a final gene set consisting of 370 genes (**S4 Table**) and we called this gene set proliferation signature.

To use proliferation signature gene set to predict proliferation rate, we apply a widely used method, ssGSEA [95], a rank-based method that computes an overexpression measure for a gene set of interest relative to all other genes in the genome, to derive a single value for a sample, which we called proliferation signature score. We use R package GSVA to apply ssGSEA with default settings [64]. The input gene expression matrix can be derived from next-generation sequencing or microarray.

To apply proliferation signature in other species, the R package Biomart [96,97] was used to obtain homologous genes of other species in this study and to map across different gene naming schemes (eg: transfer Ensemble gene id to Entrez gene id). If the mapping from human to other species is one to many, then we choose the first mapped gene. If the mapping is many to one, then we keep all mapped genes.

## Prediction of growth rates using proliferation signature

Published expression profiling data for yeasts cultured in the chemostat with controlled growth rate from Airoldi et al. (dataset1) [98], Slavov et al. (dataset2) [99], Regenberg et al. (dataset3) [14] and cancer cell lines with corresponding growth rate [66] were downloaded. For each dataset, we calculated the Pearson correlation of proliferation signature score with growth rate.

We also used another method to calculate proliferation signature score for these 3 yeast datasets. We use the sum of genes expression for all genes in the proliferation gene set to represent proliferation signature score (**S3D, S3E and S3F Fig**), the result is slightly worse than the ssGSEA method.

## Proliferation score of 2C-like mESCs and non-2C-like mESCs

RNA-seq data (GSE33923) of 2C-like mESCs are from Macfarlan et al. [67], who FACS separated 2C-like cells (high MuERVpromoter driven expression, 2C::tdTomato⁺) from non-2C-like mESCs (2C::tdTomato⁻). We calculated the proliferation signature score for each of the six samples, and used a paired t-test to control for differences between replicates.

## Brief description of experiments from other papers

In van Dijk et al. [8] *cts1Δ* histone-GFP budding yeast undergo cytokinesis to separate mother and daughter cells, but these cells remain physically attached to each other by their cell wall. Thus, starting from an initial population of single cells in G1, variation in proliferation rate can be measured by variability in histone-GFP signal in physically connected clusters of cells. Each cluster contains cells descended from the same ancestor cell.

In Dhar et al. [9] wild-type yeast were stained with TMRE, and sorted into four bins with varying TMRE signal.

In Sukumar et al. [22] pmel-1 T cell receptor (TCR) transgenic mice were injected with recombinant vaccinia virus encoding hgp100 (gp100-VV). Five days after vaccination, they isolated CD8+ T cells, stained them with the lipophilic cationic dye tetramethylrhodamine methyl ester (TMRM) (25 nm for 30 min at 37°C) and FACS-sorted the highest and lowest 7–10% of cells for subsequent RNA-seq.

## Proliferation scores of preterminal cell lineages vs terminal cell types

Preterminal cell lineage and terminal cell type pseudo-bulk RNAseq data of *C. elegans* were downloaded from Murray et al. [69], specifically, gene expression profile for terminal cell types and preterminal cell lineage is in S7 and S8 Tables in Ref. [69], annotation file for terminal cell types and preterminal cell lineage is in S2 and S4 Tables in Ref. [69]. As there are multiple time points for one terminal cell type, we only use the sample with maximum time point to represent the corresponding terminal cell type, processed data provided in this study (**S9 Table**). For each cell we calculate proliferation signature score, and a t-test was used to compare the mean proliferation signature score of all cells in each of the two groups.

## *C. elegans* scRNA-seq data analysis

*C. elegans* scRNA-seq data was provided in R package "VisCello.celegans". After loading the package, type cello() to load all data into the current environment. We calculated the proliferation signature score for all single cells, then color them by proliferation signature score in UMAP space. The calculation of proliferation signature score for single cell data is different from the calculation for bulk RNA-seq data. We just sum up the expression value of genes in the proliferation signature gene set to get a proliferation signature score. We do not use ssGSEA consider ssGSEA is a rank-based method, however most of the genes have 0 expression in this scRNA-seq data set (**S4A Fig**), which makes it not appropriate to use ssGSEA.

We binned all single cells by their embryo time. We first calculate the mean proliferation score for cells with same embryo time, then calculate Spearman correlation of this mean proliferation score with embryo time, the result is rho = -0.65 (p = $9.3 \times 10^{-19}$), the correlation of unbinned data is -0.41 (p < $2.2 \times 10^{-16}$). After excluding three special cell types germline, M cell and intestine, the result is rho = -0.73 (p = $8.6 \times 10^{-24}$), the correlation of unbinned data is -0.45 (p < $2.2 \times 10^{-16}$).

## Experimental data for *C. elegans* development

Developmental data of *C. elegans* was extracted from Fig 4 of Sulston et al. [70]. This figure is cell number (live nuclei number) change over embryo time and we use WebPlotDigitizer [100] to extract data. We use the data to plot log2 cell number change over embryo time. The difference of log2 cell number for two adjacent time points divided by the difference of time is the proliferation rate of mean of two time points.

## Supporting information

**S1 Fig. Fast proliferating subpopulations maintain higher proliferation rates than slow proliferating subpopulations for at least 3 days. FACS gating strategy for CFSE staining to get cell subpopulations with different proliferation rates. (A, B)** The gating strategy for CFSE staining to get cell subpopulations with different proliferation rates in fibroblasts **(A)** and in ESCs **(B)**. Slow, medium and fast proliferating cell subpopulations were sorted by FACS according to their CFSE signal. Then RNA-seq was performed on each of the three sub-populations. **(C, D)** FACS gating strategy for measuring the heritability of proliferation rates. The gating strategy for measuring the heritability of proliferation rates in fibroblasts **(C)** and in ESCs **(D)**. In all experiments, the laser voltage was increased so that, when sorting high and low CFSE cells, the modal CFSE signal was at least $10^3$; the voltage is not the same for the first and second CFSE sorts. **(E)** 3 Replicates of fibroblasts that similar to **Fig 1E**. **(F)** 2 Replicates of ESCs that similar to **Fig 1F**.
(TIF)

**S2 Fig. Functional pathways for which cell-to-cell heterogeneity in expression correlates with proliferation rate across cell types and species. (A)** GSEA result plot of Go preribosome genes set for ESC. **(B)** The heatmap shows the expression (z-scored read counts) of preribo-some genes in ESCs across four biological replicates of the CFSE sorting experiment. **(C)** Higher expression of Myc in both fast proliferating ESCs and fibroblasts. log2 fold change of Myc expression between fast and slow proliferating subpopulation in both ESCs and fibro-blasts, each cell type 4 replicates. **(D)** Correlated changes in the expression of ribosome biogenesis and proteasome related genes during organ development. Change of average expression of log2(TPM+1) of genes in ribosome biogenesis (Go preribosome) gene set and proteasome complex (Go proteasome complex) gene set with developmental stages across different organs in seven species [16]. Points (circle and triangle) are the mean expression of replicates, error bars represent the maximum and minimum value in the replicates.
(TIF)

**S3 Fig. Proliferation signature scores predict growth rate, using different methods of calculation, and different species. (A-C)** Using the Normalized Enrichment Score from ssGSEA to predict growth rate in three different data sets. The Pearson correlation of proliferation signature score with growth rate in are R = 0.82 (p = $8.9 \times 10^{-7}$), R = 0.73 (p = $1.3 \times 10^{-8}$) and R = 0.77 (p = $3.7 \times 10^{-3}$). **(D-F)** Similar to A-C, but using the sum of expression values for all genes in the proliferation signature gene set to calculate proliferation signature score. The Pearson correlation of proliferation signature score with growth rate are R = 0.83 (p = $7.0 \times 10^{-7}$), R = 0.73 (p = $1.6 \times 10^{-8}$) and R = 0.55 (p = $0.65 \times 10^{-2}$).
(TIF)

**S4 Fig. Lineage-specific proliferation signature scores during *C. elegans*, human and mouse development. (A)** A density histogram of counts (UMI) across 1000 randomly sampled cells; 95.6% of genes have zero reads. This causes ssGSEA to give unreliable results, so the sum of expression values method is used for calculating the proliferation signature score for single cells. **(B)** Cells are binned by embryo time, and the mean proliferation signature score for all cells not the three outlier cell types (germline, intestine and M cells). The Spearman correlation is rho = -0.73 (p = $8.7 \times 10^{-24}$) for binned data, and rho = -0.45 (p $<$ $2.2 \times 10^{-16}$) for unbinned data. **(C)** Similar to Fig 5C, but only showing cells with an age higher than 650 minutes, rho = 0.5 (p = $1.5 \times 10^{-2}$). **(D, E)** The change in cell number, and the rate of change in cell number, during development, as measured by microscopy [70]. **(F)** Change in proliferation signature score for seam cells, which form multinucleated cells through cell-fusion. **(G)**

Human single-cell gene expression data from Petropoulos et al. [72] projected onto the first two principal components and colored by proliferation signature score or developmental stages. And boxplot shows the change of proliferation signature score with developmental stages. **(H)** Similar to G, but using mouse scRNA-seq data from Deng et al. [73].
(TIF)

**S5 Fig. FACS gating strategy for TMRE staining and volcano plots for fibroblasts and ESCs sorted by CFSE or TMRE. (A, B)** The gating strategy for TMRE staining to get cell sub-populations with different mitochondrial states in fibroblasts **(A)** and in ESCs **(B)**. We use Hoechst to get cells in G0/G1, gate by CFSE to get a more uniform cell population, and separate populations with high and low TMRE signal, then do RNA-seq on each of the two subpopulations. **(C, D)** Deseq2 was used to calculate log2 fold change and adjusted p-values for CFSE sorting **(C)** and TMRE sorting **(D)**, combining biological replicates. To set the axes to be maximally informative, genes with $p < 10^{-5}$ had p set to $10^{-5}$, and those abs(log2 fold change) > 5 were truncated at -5 or +5. **(E)** PCA for RNA-seq data of ESCs sorted by TMRE. Low TMRE ESCs of replicate 1 is an outlier, so we remove replicate 1 for all analysis.
(TIF)

**S6 Fig. N-cadherin and E-cadherin levels in oligomycin and antimycin treated cells. (A)** E-cadherin staining of ESCs as positive control for E-cadherin detection, Scale bar = 80 μm. Immunostaining for E-cadherin does not show detectable levels in fibroblasts, Scale bars = 15 μm. **(B)** Western blot for N-cadherin and E-cadherin (Gapdh = loading control) in DMSO- and drug-treated fibroblasts and ESCs. Oligomycin, but not antimycin treatment led to a reduction in N-cadherin in fast fibroblasts. Double-drug treatment also showed no strong effect likely due to antimycin treatment (complex III) acting upstream of oligomycin treatment (complex V) in the electron transport chain and therefore being the dominant drug in the double-treatment.
(TIF)

**S7 Fig. The effects of Hoechst and CFSE staining on cell viability and proliferation rates.** Shown are the estimated doubling times (based on the increased in cell number after 24hrs growth) and measured viability (trypan blue) for fibroblasts **(A)** and ESCs **(B)**. The microscopy shows that stained cells maintain the correct morphology.
(TIF)

**S8 Fig. The images of CFSE staining for both fast and slow proliferating fibroblasts and ESCs.** Fibroblasts **(A)** and ESCs **(B)** were stained by CFSE and sorted into two bins: fast proliferating cells (low CFSE) and slow proliferating cells (high CFSE).
(TIF)

**S9 Fig. Images of TMRE staining for fibroblasts and ESCs, showing heterogeneity.** Brightfield and TMRE staining images for both Fibroblasts and ESCs. Two pairs of ESC colonies of similar size but showing staining heterogeneity are circled in blue.
(TIF)

**S1 Table. Viability data.**
(XLSX)

**S2 Table. Overlap of GSEA results of fibroblasts and ESCs (CFSE) and GSEA result of yeast Fitflow.**
(XLSX)

**S3 Table. Cor table prolif vs gene expression.**
(TSV)

**S4 Table. Proliferation signature combined human C. elegans yeast mouse.**
(XLSX)

**S5 Table. GSEA result of fibroblasts TMRE and ESCs TMRE.**
(XLSX)

**S6 Table. TMRE sort ANOVA.**
(XLSX)

**S7 Table. CD8T Ribosome and Proteasome table.**
(XLSX)

**S8 Table. GSEA results of Fibroblasts, ESCs and yeast sorted by CFSE or TMRE.**
(XLSX)

**S9 Table. C. elegans id single cell 20201027 max as terminal.**
(TSV)

**S10 Table. Published data used in this paper.**
(XLSX)

**S11 Table. Yeast Fit Flow Fast Slow.**
(TSV)

## Acknowledgments

We thank the CRG/UPF flow-cytometry core and CRG Genomics facilities for help with experiments, Yang Zhao for comments on the manuscript, and Elvan Boke and Aida Rodriguez for valuable advice on mitochondria drug treatments.

## Author Contributions

**Conceptualization:** Zhisheng Jiang, Serena F. Generoso, Bernhard Payer, Lucas B. Carey.

**Data curation:** Zhisheng Jiang, Serena F. Generoso, Marta Badia, Lucas B. Carey.

**Formal analysis:** Zhisheng Jiang, Marta Badia, Lucas B. Carey.

**Funding acquisition:** Bernhard Payer, Lucas B. Carey.

**Investigation:** Zhisheng Jiang, Serena F. Generoso.

**Methodology:** Zhisheng Jiang, Serena F. Generoso.

**Project administration:** Bernhard Payer, Lucas B. Carey.

**Resources:** Bernhard Payer, Lucas B. Carey.

**Supervision:** Bernhard Payer, Lucas B. Carey.

**Visualization:** Zhisheng Jiang, Serena F. Generoso.

**Writing – original draft:** Zhisheng Jiang, Serena F. Generoso, Bernhard Payer, Lucas B. Carey.

**Writing – review & editing:** Zhisheng Jiang, Serena F. Generoso, Bernhard Payer.

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
