## [Decision Letter · Decision Letter 0]

29 May 2021

Dear Dr. Carey,

Thank you very much for submitting your manuscript "A conserved expression signature predicts growth rate and reveals cell & lineage-specific differences" for consideration at PLOS Computational Biology. As with all papers reviewed by the journal, your manuscript was reviewed by members of the editorial board and by several independent reviewers. The reviewers appreciated the attention to an important topic. Based on the reviews, we are likely to accept this manuscript for publication, providing that you modify the manuscript according to the review recommendations.

Sincerely,

Elana J. Fertig, PhD

Associate Editor

PLOS Computational Biology

Mark Alber

Deputy Editor

PLOS Computational Biology

[LINK]

Reviewer's Responses to Questions

**Comments to the Authors:**

Reviewer #1: Jiang et al present a clever experimental approach to measuring growth rate-dependent gene expression, and find a gene-expression profile that predicts growth rate across a range of species and cell types. The experimental approach is to label cells with a stable, cell-permeable dye, sort the cells to get a homogeneously fluorescent starting population, and then sort cells at a later time for RNA-seq according to how much fluorescence remains (because fluorescence decreases as cells grow and divide). Overall, the experiments are rigorously performed and the predictive power is impressive, especially with cross-species validation. The paper will be an important contribution to the study of cell-to-cell heterogeneity and growth control. However, I do have two main recommendations to strengthen it:

1) The manuscript could do a better job of placing this work in the context of the existing literature. The work of Terry Hwa and colleagues on the connection between translational capacity and growth is not cited. Recent work identifying semi-heritable cell-to-cell heterogeneity in mammalian cells using RNA-seq is not cited (Raj lab and Tanay lab). Wytock & Motter 2019 is cited, but other work on predicting growth rate from gene expression is not (e.g., doi:10.1371/journal.pone.0075320 which discusses ribosomal protein expression).

2) The computational/prediction method should be more completely discussed in the main text and the method should be more explicitly justified. For example, the cutoff for growth-correlated genes of rho = 1 seems a bit confusing at first (line 255), until one considers that this is a rank (Spearman) correlation and there are only three bins of growth rate (cells sorted with high, medium or low fluorescence). So rho = 1 just means growth rate increases monotonically with growth. This seems sensible enough, but it does not consider measurement noise the way a proper statistical model would, so this decision should be defended. (And because of experimental noise, it also would likely cause problems if, say, another experiment used 5 bins because the cutoff would likely be too stringent.) The addition of ribosome-biogenesis and proteasome gene sets to the genes that contribute to the proliferation signature score is also somewhat arbitrary (or at least one could imagine different decisions being made). So some kind of analysis of how the proliferation score is sensitive (or not) to these decisions is warranted. The performance of the proliferation score is well validated by using data from divergent organisms, but that's not to say it could not be improved. Likewise, how the gene-expression signature is transferred from one organism to another is not fully described. The Methods section explains that homologs were identified using the R package Biomart, but more detail is needed. Are these 1:1 orthologs? orthologs + paralogs? It is also not entirely clear how the C. elegans profileration predictions were tested against data. It seems that the comparison is only based on the rate of increase of cell number through development. Are there C. elegans data on the actual rates of division of particular cells that could be used as well?

Reviewer #2: This paper used a fluorescence-activated cell sorting (FACS) based method to sort fast- and slow-proliferating subpopulations in mammalian cells and reports a gene expression signature, which can be used to predict the proliferation rates across eukaryotic species and cell types. This signature includes expression of house-keeping genes such as ribosome and proteasome, which are often expressed at high levels and hence can be reliably measured even using methods with low sampling efficiency (such as single-cell RNA-seq). This proliferation model is more robust than other models currently available and it has the potential to be widely used, particularly in the field of single-cell transcriptomics. Additionally, the authors found that slow-proliferating ESCs display a more naïve pluripotent expression profile than fast-proliferating ESCs. This finding is counter-intuitive and should be of interest to the stem cell field. Overall, the paper is well written, the data are well presented and supports the main collusion.

The weaker part of the paper is the cell-type specific correlation between mitochondria function and proliferation rates, where the authors seem to have made efforts towards gaining mechanistic insights behind the correlation. But the data seem piecemeal and it is unclear what message these data really deliver, particularly with seemingly contradicting data unexplained (see specific comments). Compared to other important findings in the paper, this part could perhaps be reduced to a more coherent and concise figure.

Specific comments:

1. What effect does protein degradation have on the intensity of CFSE dye? The authors comment that protein degradation should not have a huge effect but it should be shown with analytical/experimental evidence, especially given the finding that proteasome genes are differentially expressed between the two sorted populations. This may be tested experimentally using proteasome inhibitors.

2. Given that the entire paper is built on sorting using CFSE dye, an orthogonal method should be used to validate that the sorted slow proliferating cells are indeed proliferating more slowly. I envision that a simple growth curve should be sufficient.

3. Line 729, why only the genes with “mean(Cor(gene(i), pr(i))) = 1” are chosen as “proliferation correlated genes”? The genes with correlation value of -1 should be also highly correlated and equally important as the value 1 genes.

4. It would be useful to know which (groups of) genes in the selected gene sets contribute more for predicting the proliferation rate.

5. Contradicting statement on the relation between % S-phase cells and proliferation rates. Line 178, higher S phase fraction is used to support that fast proliferating subpopulations maintain higher proliferation rates (Figure 1 E, F). This conclusion can be also found in Line 422-423. However, in Line 293-296, for explaining no correlation between PCNA or Ki67 expression with proliferating rate, authors mentioned that “these markers are measuring the fraction of the population that is in S-phase or is not in G0…”

6. Magnitude of the proliferation score. I noticed that the proliferation score varies from 10-1 level (yeast, Figure 4B) to 102 level (human, Figure 5G). I am also curious that sometimes this score become negative. What does this mean? Can the authors comment on the condition where this score might be compared?

7. Antimycin and oligomycin are both mitochondria respiration inhibitors, yet the two have opposite effects on fast vs. slow fibroblasts (in Fig. 7C antimycin inhibits proliferation of slow FIB greater than fast FIB, but in Fig. 7F, oligomycin has a stronger effect on fast FIB). The authors should explain the discrepancy and what it really means.

8. Figure 7C, the label “Drug effect log2(DMSO/drug)” is ambiguous. What is being measured should be clearly stated in the figure or caption. Figure 7E is dim and hard to see. The cadherin staining should be quantified.

9. Figure S6B, for the same protein, experiments group and the control (e.g. ESC as the positive control for cadherin) should be on the same gel.

Other minor issues:

I. In main text, abbreviation “FACS” first appeared at Line 119 but the abbreviation was introduced at Line 160. In Figure 1(E, F) caption, abbreviation “BrdU” was used but introduced later in Materials AND Methods (Line 586).

II. The metric prefix “micro” was written as “u” instead of “μ” at Line 560 and 603, which is not a SI recommendation.

III. It’s recommended to leave a space between a number and its unit, e.g., at Line 560, 572, 586, 603.

In Line 592, there should be a space between “0.5%” and “Triton”.

IV. Line 679, “viability” was marked bold for no obvious reason.

V. Line 442 and 444, Figure 7E and S6A-C are referenced. However, Figure S6 has no panel C.

VI. Figure 4A, the gene name “KI67” in the legend is inconsistent with the “Ki67” in the caption and main text.

Reviewer #3: This is a well-written manuscript with really fantastic results.

There is strong experimental and computational work, so it will be quite well-received by PLoS Computational Biology readers. Indeed, the results the authors have shown “a nearly universal signature that is strongly associated with proliferation” will be of interest to a broad reader base spanning across biological and computational fields.

I also appreciated that, in applying their approach across organisms/data, the authors demonstrated the “overfitting” used by other methods of proliferation estimation that were based on single-species data. This level of cross-species validation is truly remarkable.

That said, there were a couple of points that would have greatly increased my comprehension of the results from an initial read that I recommend authors and editors consider. (In addition there are a few minor points/typos.)

1) I found it a bit disappointing that, in my view, that details of the critical achievement of the work: the establishment of a proliferation score, was relegated to the methods section.

I’m a mathematical modeler, so at least from my point of view the proliferation score itself is a result of the paper and merits greater discussion in the main text.

Although it relies on ssGSEA, an established method, some details within the manuscript of how the score itself is calculated would have been appreciated.

2) To that end, reading the methods it appears that the authors consider only the genes with a correlation of 1 to proliferation in calculating their score. While clearly, this results in a highly predictive and not-overfit model, this criteria seemed quite strict.

Could the authors elaborate if, perhaps within a species, there are genes that are close to perfect correlation that would increase their predictive power? Or once you move outside perfect correlation do you encounter overfitting?

3) I am not an experimentalist, but have worked with CFSE data as a modeler. So I had some questions about these experiments.

Were the cells at the start were synchronized in some way? It seemed to me that heterogeneity in the cell cycle at the start of the experiment would lead to complications in identifying the “fast” from “slow” cells as you might capture who was “lucky” enough to be at the right point in the cell cycle.

Again, I am familiar with mathematical approaches to estimating proliferation rates from CFSE data (such as described in https://www.ncbi.nlm.nih.gov/pmc/articles/PMC3196292/). The CFSE data in this manuscript (figure 1) is far “cleaner” than I am used to seeing from these previously published studies. If there non-standard experimental procedures which produced this cleaned data, I would recommend they be highlighted in the main text.

Minor Points:

• Figure 2: Panels (B) and (C), the caption should define what the “grey” regions around the linear fit. (Is this a 90% confidence band? 95% confidence band?)

• Line 109: Is the first time in the main text that ESCs are used, I would recommend spelling this out.

• Line 139: “cause” should be “causes”

• Line 732: I was I was confused by what was meant by “proliferation rate” in the correlation? Is this just the normally calculated doubling time of a sample?

• Lines 793: The sentence beginning “We just sum up…” is quite long and hard to parse. I recommend putting a period after “proliferation signature score” and making a new sentence beginning “We do not use ssGSEA…”

**Have the authors made all data and (if applicable) computational code underlying the findings in their manuscript fully available?**

Reviewer #1: Yes

Reviewer #2: Yes

Reviewer #3: Yes

PLOS authors have the option to publish the peer review history of their article (what does this mean?). If published, this will include your full peer review and any attached files.

Reviewer #1: No

Reviewer #2: No

Reviewer #3: No

Figure Files:

Data Requirements:

Reproducibility:

References:

---

## [Decision Letter · Decision Letter 1]

21 Oct 2021

Dear Dr Payer,

We are pleased to inform you that your manuscript 'A conserved expression signature predicts growth rate and reveals cell & lineage-specific differences' has been provisionally accepted for publication in PLOS Computational Biology.

Best regards,

Elana J. Fertig, PhD

Associate Editor

PLOS Computational Biology

Mark Alber

Deputy Editor

PLOS Computational Biology

Reviewer's Responses to Questions

**Comments to the Authors:**

Reviewer #1: The authors have responded well to suggestions from the previous round of review. I'm still not completely convinced that using a rank correlation threshold of 1 is the best way to go, because it is very specific to this work's use of 3 bins. I would still prefer to see a statistical treatment that models measurement noise. Nonetheless, the authors' results and conclusions are convincing, so this is more a comment about the general utility of this approach to others who subsequently perform similar analyses and not a comment on the rigor of the present work.

Reviewer #2: The authors have addressed all my questions.

Reviewer #3: The authors have satisfied my questions and concerns.

I believe the paper is ready for publication.

**Have the authors made all data and (if applicable) computational code underlying the findings in their manuscript fully available?**

Reviewer #1: Yes

Reviewer #2: Yes

Reviewer #3: Yes

PLOS authors have the option to publish the peer review history of their article (what does this mean?). If published, this will include your full peer review and any attached files.

Reviewer #1: No

Reviewer #2: No

Reviewer #3: No

---

## [Editor Report · Acceptance letter]

5 Nov 2021

PCOMPBIOL-D-21-00616R1 

A conserved expression signature predicts growth rate and reveals cell & lineage-specific differences

Dear Dr Payer,

I am pleased to inform you that your manuscript has been formally accepted for publication in PLOS Computational Biology. Your manuscript is now with our production department and you will be notified of the publication date in due course.

With kind regards,

Livia Horvath
